# LEARNING CONTROL BY ITERATIVE INVERSION

## ABSTRACT

We formulate learning for control as an *inverse problem* – inverting a dynamical system to give the actions which yield desired behavior. The key challenge in this formulation is a *distribution shift* in the inputs to the function to be inverted – the learning agent can only observe the forward mapping (its actions' consequences) on trajectories that it can execute, yet must learn the inverse mapping for inputs-outputs that correspond to a different, desired behavior. We propose a general recipe for inverse problems with a distribution shift that we term *iterative inversion* – learn the inverse mapping under the current input distribution (policy), then use it on the desired output samples to obtain a new input distribution, and repeat. As we show, iterative inversion can converge to the desired inverse mapping, but under rather strict conditions on the mapping itself.

We next apply iterative inversion to learn control. Our input is a set of demonstrations of desired behavior, given as video embeddings of trajectories (without actions), and our method iteratively learns to imitate trajectories generated by the current policy, perturbed by random exploration noise. We find that constantly adding the demonstrated trajectory embeddings *as input* to the policy when generating trajectories to imitate, a-la iterative inversion, we effectively steer the learning towards the desired trajectory distribution. To the best of our knowledge, this is the first exploration of learning control from the viewpoint of inverse problems, and the main advantage of our approach is simplicity – it does not require rewards, and only employs supervised learning, which can be easily scaled to use state-of-the-art trajectory embedding techniques and policy representations. Indeed, with a VQ-VAE embedding, and a transformer-based policy, we demonstrate non-trivial continuous control on several tasks. Further, we report an improved performance on imitating diverse behaviors compared to reward based methods.

## 1 INTRODUCTION

The control of dynamical systems is fundamental to various disciplines, such as robotics and automation. Consider the following trajectory tracking problem. Given some deterministic but unknown actuated dynamical system,

$$s_{t+1} = f(s_t, a_t), \tag{1}$$

where $s$ is the state, and $a$ is an actuation, and some reference trajectory, $s_0, \ldots, s_T$, we seek actions that drive the system in a similar trajectory to the reference.

For system that are 'simple' enough, e.g., linear, or low dimensional, classical control theory (Bertsekas, 1995) offers principled and well-established system identification and control solutions. However, for several decades, this problem has captured the interest of the machine learning community, where the prospect is scaling up to high-dimensional systems with complex dynamics by exploiting patterns in the system (Mnih et al., 2015; Lillicrap et al., 2015; Bellemare et al., 2020).

In reinforcement learning (RL), learning is driven by a manually specified *reward* signal $r(s, a)$. While this paradigm has recently yielded impressive results, defining a reward signal can be difficult for certain tasks, especially when high-dimensional observations such as images are involved. An alternative to RL is inverse RL (IRL), where a reward is not manually specified. Instead, IRL algorithms *learn* an implicit reward function that, when plugged into an RL algorithm in an inner loop, yields a trajectory similar to the reference. The signal driving IRL algorithms is a *similarity metric between trajectories*, which can be manually defined, or learned (Ho & Ermon, 2016).

We propose a different approach to learning control, which does not require explicit nor implicit reward functions, and also does not require a similarity metric between trajectories. Our main idea is that Equation (1) prescribes a mapping $\mathcal{F}$ from a sequence of actions to a sequence of states,

$$s_0, \ldots, s_T = \mathcal{F}(a_0, \ldots, a_{T-1}). \qquad (2)$$

The control learning problem can therefore be framed as finding the *inverse function*, $\mathcal{F}^{-1}$, without knowing $\mathcal{F}$, but with the possibility of evaluating $\mathcal{F}$ on particular action sequences (a.k.a. roll-outs).

Learning the inverse function $\mathcal{F}^{-1}$ using regression can be easy if one has samples of action sequences and corresponding state sequences, and a distance measure over actions. However, in our setting, we do not know the action sequences that correspond to the desired reference trajectories. Interestingly, for some mappings $\mathcal{F}$, an iterative regression technique can be used to find $\mathcal{F}^{-1}$. In this scheme, which we term *Iterative Inversion* (IT-IN), we start from arbitrary action sequences, collect their corresponding state trajectories, and regress to learn an inverse. We then apply this inverse on the reference trajectories to obtain new action sequences, and repeat. We show that with linear regression, iterative inversion will converge under quite restrictive criteria on $\mathcal{F}$, such as being strictly monotone and with a bounded ratio of derivatives. Nevertheless, our result shows that for some systems, a controller can be found without a reward function, nor a distance measure on states.

We then apply iterative inversion to several continuous control problems. In our setting, the desired behavior is expressed through a video embedding of a desired trajectory, using a VQ-VAE (Van Den Oord et al., 2017), and a deep network policy maps this embedding and a state history to the next action. The agent generates trajectories from the system using its current policy, *given the desired embeddings as input*, and subsequently learns to imitate its own trajectories, conditioned on their own embeddings. Interestingly, we find that when iterating this procedure, the input of the desired trajectories' embeddings *steers* the learning towards the desired behavior, as in iterative inversion.

Given the strict conditions for convergence of iterative inversion, there is no a-priori reason to expect that our method will work for complex non-linear systems and expressive policies. Curiously, however, we report convergence on all the scenarios we tested, and furthermore, the resulting policy generalized well to imitating trajectories that were not seen in its 'steering' training set. This surprising observation suggests that IT-IN may offer a simple supervised learning-based alternative to methods such as RL and IRL, with several potential benefits such as a reward-less formulation, and the simplicity and stability of the (iterated) supervised learning loss function. Furthermore, on experiments where the desired behaviors are abundant and diverse, we report that IT-IN outperforms reward-based methods, even with an accurate state-based reward.

## 2  RELATED WORK

In learning from demonstration (Argall et al., 2009), it is typically assumed that the demonstration contain both the states and actions, and therefore supervised learning can be directly applied, either by behavioral cloning (Pomerleau, 1988) or interactive methods such as DAgger (Ross et al., 2011). In our work, we assume that only states are observed in the demonstrations, precluding straightforward supervised learning. Inverse RL is a similar problem to ours, and methods such as apprenticeship learning (Abbeel & Ng, 2004) or generative adversarial imitation learning (Ho & Ermon, 2016) simultaneously train a critic that discriminates between the data trajectories and the policy trajectories (a classification problem), and a policy that confuses the critic as best as possible (an RL problem). It is shown that this procedure will converge to a policy that visits the same states as the data. While works such as (Fu et al., 2019; Ding et al., 2019) considered a goal-conditioned IRL setting, we are not aware of IRL methods that can be conditioned on a more expressive description than a target goal state, such as a complete trajectory embedding, as we explore here. In addition, our approach avoids the need of training a critic, as in Ding et al. (2019), or training an RL agent in an inner loop.

Most related to our work, Ghosh et al. (2019) proposed goal conditioned supervised learning (GCSL). In GCSL, the agent iteratively executes random trajectories, and uses them as direct supervision for training a goal-conditioned policy, where observed states in the trajectory are substituted as goals. The desired goals are also input to the policy when generating the random trajectories. In comparison to GCSL, we do not consider tasks of only reaching goal states, but tasks where the *whole trajectory is important*. This significantly increases the diversity of possible tasks, and thereby increases the difficulty of the problem. In addition, the theoretical analysis of Ghosh et al. (2019)

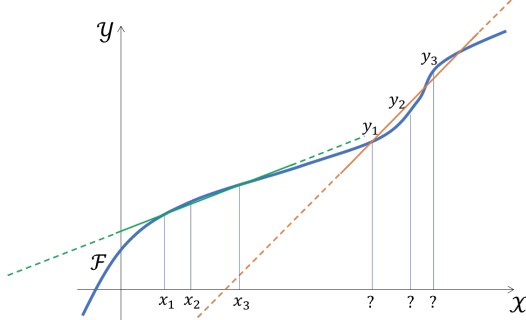

Figure 1: Learning an inverse function under a distribution shift. We wish to learn the inverse function over outputs $y_1, \ldots, y_M$, using linear least squares, having matching inputs-outputs for $x_1, \ldots, x_M$.

showed convergence under an assumption that all goal states have a probability of being visited under the initial policy. The analysis we show for iterated inversion, and the fact that this assumption almost never holds in practice (except, e.g., in offline RL, where the data is already 'explorative' enough Emmons et al. 2021), suggest that the practical success of GCSL is less obvious than as the theory of Ghosh et al. (2019) predicts.

In self-supervised RL, the agent is not given reward, and uses its own experience to explore the environment, typically by training a goal-conditioned policy, and proposing to it goals that are novel in some measure (Ecoffet et al., 2019; Hazan et al., 2019; Sekar et al., 2020; Endrawis et al., 2021; Mendonca et al., 2021). The space of all trajectories is much larger than the space of all states, and we are not aware of methods that demonstrably explore such a space. For this reason, in our approach we steer the exploration towards a set of desired trajectories.

Very recently, in their work on video pretraining, Baker et al. (2022) also used a transformer to learn an inverse model conditioned on a video. Importantly, Baker et al. (2022) collected human-labelled data to train their inverse model on desired behavior trajectories. The main point in our work is a self-supervised learning paradigm that automatically steers data collection to the desired behavior.

Finally, we mention the extensive literature on deep learning based solutions for inverse problems (Lucas et al., 2018; Kamyab et al., 2021). In many studies, the forward mapping is known, and differentiated to iteratively optimize reconstruction (Xia et al., 2022). In contrast, blind inversion methods learn the inverse mapping directly from data (Kulkarni et al., 2016). To the best of our knowledge, the formulation of learning control as an inverse problem, and iterative inversion as a solution for the resulting distribution shift problem, are novel.

## 3   ITERATIVE INVERSION

In this section we describe a general problem of learning an inverse function under a distribution shift, and present the iterative inversion algorithm. We then analyse the convergence of iterative inversion in several simplified settings. In the proceeding, we will apply iterative inversion to learning control.

Let $\mathcal{F} : \mathcal{X} \to \mathcal{Y}$ be a bijective function. We are given a set of $M$ desired outputs $y_1, \ldots, y_M \in \mathcal{Y}$, and an arbitrary set of $M$ initial inputs $x_1, \ldots, x_M \in \mathcal{X}$. We assume that $\mathcal{F}$ is not known, but we are allowed to observe $\mathcal{F}(x)$ for any $x \in \mathcal{X}$ that we choose during our calculations. Our goal is to find a function $\mathcal{G} : \mathcal{Y} \to \mathcal{X}$ such that for any desired output $y_i$, we have $\mathcal{G}(y_i) = \mathcal{F}^{-1}(y_i)$.

More specifically, we will adopt a parametric setting, and search for a parametric function $\mathcal{G}_\theta$, where $\theta \in \Theta$ is a parameter vector, that minimizes the average loss:

$$\min_{\theta \in \Theta} \frac{1}{M} \sum_{i=1}^{M} \mathcal{L}(\mathcal{G}_\theta(y_i), \mathcal{F}^{-1}(y_i)). \tag{3}$$

For example, $\mathcal{G}_\theta$ could represent the space of linear functions $\mathcal{G}_\theta(y) = \theta^T y + \theta_0$, and $\mathcal{L}$ could be the squared error between inputs, $\mathcal{L}(x, x') = (x - x')^2$. This example, which is depicted in Figure 1 for the 1-dimensional case $\mathcal{X} = \mathcal{Y} = \mathbb{R}$, corresponds to a linear least squares fit of the inverse function. As can be seen, the challenge in this problem arises from the mismatch between the distributions of the desired outputs and initial inputs.

The iterative inversion algorithm, proposed in Algorithm 1, seeks to solve problem (3) iteratively.

---

**Algorithm 1** Iterative Inversion

---

**Require:** Desired outputs $y_1, \ldots, y_M \in \mathcal{Y}$, loss function $\mathcal{L} : \mathcal{X} \times \mathcal{X} \to \mathbb{R}$, initial parameter $\theta_0$.
1: **for** $n = 0, 1, 2, \ldots$ **do**
2:     Calculate current inputs: $x_1^n, \ldots, x_M^n = \mathcal{G}_{\theta_n}(y_1), \ldots, \mathcal{G}_{\theta_n}(y_n)$
3:     Calculate current outputs: $y_1^n, \ldots, y_M^n = \mathcal{F}(x_1^n), \ldots, \mathcal{F}(x_M^n)$
4:     Regression

$$\theta_{n+1} = \arg\min_{\theta \in \Theta} \frac{1}{M} \sum_{i=1}^{M} \mathcal{L}(\mathcal{G}_\theta(y_i^n), x_i^n)$$

5: **end for**

---

We next investigate when, and why, should iterative inversion produce an effective solution for (3). We restrict ourselves to a linear function class for $\mathcal{G}_\theta$, and the squared loss. We report convergence results for different classes of functions $\mathcal{F}$.

Denote $X^n \equiv (x_1^n, \ldots, x_M^n)^T \in \mathbb{R}^{M \times dim(\mathcal{X})}$, $\mathcal{F}(X^n) \equiv (\mathcal{F}(x_1^n), \ldots, \mathcal{F}(x_M^n))^T \in \mathbb{R}^{M \times dim(\mathcal{Y})}$ as the input and output matrices, $\overline{X^n} \equiv \sum_{i=1}^{M} x_i^n / M \in \mathbb{R}^{dim(\mathcal{X})}$ and $\overline{Y} \equiv \sum_{i=1}^{M} y_i / M$, $\overline{\mathcal{F}(X^n)} \equiv \sum_{i=1}^{M} \mathcal{F}(x_i^n) / M \in \mathbb{R}^{dim(\mathcal{Y})}$ as the inputs, desired outputs, and current outputs means, $(\cdot)^\dagger$ the Moore-Penrose pseudoinverse operator, and $\mathcal{F}^{-1}$ the ground-truth inverse function.

We start with the simple case of a linear $\mathcal{F}$. As is clear from Figure 1, the distribution shift is not a problem in this case, and iterative inversion converges in a single iteration.

**Theorem 1.** *If $\mathcal{F}$ is a linear function and $rank(\mathcal{F}(X^0) - \overline{\mathcal{F}(X_0)}) = dim(\mathcal{Y})$ then Algorithm 1 converges in one iteration, i.e., $y_1^1, \ldots, y_M^1 = y_1, \ldots, y_M$.*

Iterative Inversion can be interpreted as a variant of the classic Newton's method (Ortega & Rheinboldt, 2000), where we replace the unknown Jacobian $J$ of $\mathcal{F}$ with a linear approximation using the current input-output pairs, and the evaluation of $\mathcal{F}$ with the mean of the current outputs. Recall that Newton's method seeks to find the root $x^*$ of a function $r(x) = \mathcal{F}(x) - y$ using the iterative update rule $x^{n+1} = x^n + (y - \mathcal{F}(x^n))[J(x^n)]^{-1}$, where $[J(x^n)]^{-1}$ is the Jacobian inverse of $\mathcal{F}$ at $x^n$. Iterative Inversion, similarly, applies the following updating rule, as proved in Appendix A.1,

$$\overline{X^{n+1}} = \overline{X^n} + \left( \overline{Y} - \overline{\mathcal{F}(X^n)} \right) \tilde{J}_n^{-1}, \tag{4}$$

where $\tilde{J}_n^{-1} \equiv (\mathcal{F}(X^n) - \overline{\mathcal{F}(X^n)})^\dagger (X^n - \overline{X^n})$ is the Jacobian of $\mathcal{G}_{\theta_{n+1}}$, the linear regressor plane from $\mathcal{F}(x)$ to $x$ at $x_1^n, \ldots, x_M^n$, which can be considered to be an approximation of $[J(\overline{X^n})]^{-1}$. When the approximations $\tilde{J}_n^{-1} \approx [J(\overline{X^n})]^{-1}$ and $\overline{\mathcal{F}(X^n)} \approx \mathcal{F}(\overline{X^n})$ are accurate, Iterative Inversion coincides with Newton's method, and enjoys similar convergence properties, as we establish next.

**Assumption 1.** *$\mathcal{F} : \mathbb{R}^K \to \mathbb{R}^K$ is bijective, and $\mathcal{F}$ and $\mathcal{F}^{-1}$ are both continuously differentiable.*

Denote $J(x)$ the Jacobian matrix of $\mathcal{F}$ at $x \in \mathcal{X}$, and $J^{-1}(x) \equiv [J(x)]^{-1}$ the inverse matrix of $J(x)$ and the Jacobian of $\mathcal{F}^{-1}$ at $\mathcal{F}(x) \in \mathcal{Y}$, under Assumption 1. Also denote $\| \cdot \|$ to be any induced norm (Horn & Johnson, 2012). We assume that the derivatives of $\mathcal{F}$ and $\mathcal{F}^{-1}$ are bounded, as follows.

**Assumption 2.** *$\|J(x_1) - J(x_2)\| \leq \gamma$, $\|J(x)\| \leq \zeta$ and $\|J^{-1}(x)\| \leq \beta \ \forall x_1, x_2, x \in \mathbb{R}^K$.*

Further assume that at every iteration $n$, the approximations $\tilde{J}_n^{-1}$ and $\overline{\mathcal{F}(X^n)}$ are accurate enough.

**Assumption 3.** *$\forall n$: $\|\overline{\mathcal{F}(X^n)} - \mathcal{F}(\overline{X^n})\| \leq \lambda$ and $\tilde{J}_n^{-1} = J^{-1}(\overline{X^n})(I + \Delta_n)$, $\|\Delta_n\| \leq \delta < 1/\zeta\beta$.*

Assumption 3 may hold, for example, when the inputs $x_1^n, \ldots, x_M^n$ are distributed densely, relative to the curvature of $\mathcal{F}$, and evenly, such that the regression problem in Algorithm 1 is well-conditioned. The requirement $\delta < 1/\zeta\beta$ is set to ensure that $\tilde{J}_n^{-1}$ is non-singular.

**Theorem 2.** *Suppose Assumptions 1, 2 and 3 hold. Let $\mu \equiv \frac{\zeta^2 \beta \delta}{1 - \zeta\beta\delta}$ and assume $\beta(1+\delta)(\gamma + \mu) < 1$. Let $\rho \equiv \frac{2\lambda\beta(1+\delta)(\mu+\zeta)}{1 - \beta(1+\delta)(\mu+\gamma)}$. Then for every $\epsilon > 0$ there exists $k < \infty$ such that $\|\overline{\mathcal{F}(X^k)} - \overline{Y}\| \leq \rho + \epsilon$.*

The proof for Theorem 2 builds on the analysis of Newton's method to show that IT-IN is an iterated contraction, and is reported in Section A.3 of the supplementary material. In Theorem 2, the term $\rho$ can be interpreted as the radius of the ball centered at $\overline{Y}$ that the sequence convergences to. To get some intuition about Theorem 2, consider the 1-dimensional case $\mathcal{F} : \mathbb{R} \to \mathbb{R}$, where the approximations in Assumption 3 are perfect, i.e., $\lambda = \delta = \mu = \rho = 0$. Then, the condition for convergence is $\beta(1 + \delta)(\gamma + \mu) = \beta\gamma < 1 \implies \frac{\max |\mathcal{F}'(x)|}{\min |\mathcal{F}'(x)|} < 2$, which can be interpreted as a 'close to linear' $\mathcal{F}$. The conditions in Theorem 2 can therefore be intuitively interpreted as $\mathcal{F}$ being 'close to linear' globally, and the linear approximation being accurate locally.

In Appendix A.4, we provide additional convergence results that use a different analysis technique for the simple case $\mathcal{F} : \mathbb{R} \to \mathbb{R}$. These results do not require Assumption 3, but still require a condition similar to $\frac{\max |\mathcal{F}'(x)|}{\min |\mathcal{F}'(x)|} < 2$, and show a linear convergence rate. We further remark that a quadratic convergence rate is known for Newton's method when the initial iterate is close to optimal; we believe that similar results can be shown for IT-IN as well. Here, however, we focused on the case of an arbitrary initial iterate, similarly to the experiments we shall describe in the sequel.

## 4 ITERATIVE INVERSION FOR LEARNING CONTROL

In this section, we apply iterative inversion for learning control. We first present our problem formulation, and then propose an IT-IN algorithm.

We follow a standard RL formulation. Let $S$ denote the state space, $A$ denote the action space, and consider the dynamical system in Equation 1. We assume, for simplicity, that the initial state $s_0$ is fixed, and that the time horizon is $T$.[1] Given a state-action trajectory $\tau = s_0, a_0, \ldots, s_{T-1}, a_{T-1}, s_T \in \Omega$, where $\Omega$ denotes the $T$-step trajectory space, we denote by $\tau_s \in \Omega_s$ its state component and by $\tau_a \in \Omega_a$ its action component, i.e., $\tau_s = s_0, \ldots, s_T, \tau_a = a_0, \ldots, a_{T-1}$, and $\Omega = \Omega_s \times \Omega_a$. We will henceforth refer to $\tau_s$ as a state trajectory and to $\tau_a$ as an action trajectory. Let $\mathcal{F}$ denote the mapping from an action trajectory to the resulting state trajectory, as given by Equation (2).

For presenting our control learning problem, we will assume that $\mathcal{F}$ is bijective, and therefore $\mathcal{F}^{-1}$ is well defined. We emphasize, however, that *our algorithm makes no explicit use of $\mathcal{F}^{-1}$*, and our empirical results are demonstrated on problems where this assumption does not hold.

We represent a state trajectory using an embedding function $z = \mathcal{Z}(\tau_s) \in Z$, and we term $z$ the *intent*. Note that $z$, by definition, can contain partial information about $\tau_s$, such as the goal state (Ghosh et al., 2019). In all the experiments reported in the sequel, we generated intents by feeding a rendered video of the state trajectory into a VQ-VAE encoder, which we found to be simple and well performing.

Consider a state-action trajectory $\tau$, with a corresponding intent $\mathcal{Z}(\tau_s)$. We would like to learn a policy that reconstructs the intent into its corresponding action trajectory $\tau_a$. Let $H_t$ denote the space of $t$-length state-action histories, and a policy $\pi_t : Z \times H_t \to A$. With a slight abuse of notation, we denote by $\pi(z) \in \Omega_a$ the action trajectory that is obtained when applying $\pi_t$ sequentially for $T$ time steps (i.e., a rollout). Similarly to the problem in Section 3, our goal is to learn a policy such that $\pi(\mathcal{Z}(\tau_s)) = \mathcal{F}^{-1}(\tau_s)$. More specifically, let $\mathcal{L} : \Omega_a \times \Omega_a \to \mathbb{R}$ be a loss function between action trajectories, and let $P(\tau_s)$ denote a distribution over desired state trajectories, we seek a policy $\pi_\theta$ parametrized by $\theta \in \Theta$ that minimizes the average loss:

$$\min_{\theta \in \Theta} \mathbb{E}_{\tau_s \sim P} \left[ \mathcal{L} \left( \pi_\theta(\mathcal{Z}(\tau_s)), \mathcal{F}^{-1}(\tau_s) \right) \right]. \tag{5}$$

In our approach we assume that $P(\tau_s)$ is not known, but we are given a set $D_{\text{steer}}$ of $M$ intents, $z_1, \ldots, z_M$, where $z_i = \mathcal{Z}(\tau_s^i)$, and $\tau_s^i$ are drawn i.i.d. from $P(\tau_s)$. Henceforth, we will refer to $D_{\text{steer}}$ as the *steering dataset*, as it should act to steer the learning of the inverse mapping towards the desired trajectory distribution $P(\tau_s)$.

It is worth relating Problem (5) to the general inverse problem in Section 3, and what we referred to as the distribution shift problem. Initially, the policy is not expected to be able to produce state-action trajectories that match the state trajectories in $D_{\text{steer}}$, but only trajectories that are output by the initial (typically random) policy. While these initial trajectories could be used for imitation learning,

---

[1] A varying time horizon can be handled as an additional input to $\mathcal{F}$.

yielding an intent-conditioned policy, there is no reason to expect that this policy will be any good for intents in $D_{\text{steer}}$, which are out-of-distribution with respect to the policy's training data.

We now propose a method for solving Problem (5) based on iterative inversion, as detailed in Algorithm 2. There are four notable differences from the iterative inversion method in Algorithm 1. First, we operate on batches of size $N$ instead of on the whole steering data (of size $M$), for computational efficiency. Second, we sample a batch of intents from a mixture of the steering dataset and the intents calculated for rollouts in the previous iteration. We found that this helps stabilize the algorithm. Third, we add random exploration noise to the policy when performing the rollouts, which we found to be necessary (see Sec. 5). Fourth, we used a replay buffer for the supervised learning part of the algorithm, also for improved stability. For $\mathcal{L}$, we used the MSE between action trajectories, and for the optimization in line 7, we perform several epochs of gradient-based optimization using Adam (Kingma & Ba, 2014), keeping the state history input to $\pi_\theta(\hat{z})$ fixed as $\tau_s$ when computing the gradient. The size of the replay buffer was set to $K \times N$.

---

**Algorithm 2** Iterative Inversion for Learning Control

---

**Require:** Steering data $D_{\text{steer}}$, exploration noise parameter $\eta$, steering ratio $\alpha \in [0, 1]$, batch size $N$
 1: Initialize $D_{\text{prev}} = D_{\text{steer}}$, $\theta_0$ arbitrary
 2: **for** $n = 0, 1, 2, \ldots$ **do**
 3:     Sample $\alpha N$ intents from $D_{\text{steer}}$ and $(1 - \alpha)N$ intents from $D_{\text{prev}}$, yielding $z^1, \ldots, z^N$
 4:     Perform $N$ rollouts $\tau^1, \ldots, \tau^N$ using policy $\pi_{\theta_n}$ with input intents $z^1, \ldots, z^N$, adding exploration noise $\eta$
 5:     Compute intents for the rollouts $\hat{z}^i = \mathcal{Z}(\tau_s^i)$, $i \in 1, \ldots, N$
 6:     Add intents and trajectories $\{\hat{z}^i, \tau^i\}$ to Replay Buffer
 7:     Train $\pi_{\theta_{n+1}}$ by supervised learning: $\theta_{n+1} = \arg\min_{\theta \in \Theta} \sum_{\{\hat{z}, \tau\} \in \text{Replay Buffer}} [\mathcal{L}(\pi_\theta(\hat{z}), \tau_a)]$
 8:     Set $D_{\text{prev}} = \left\{\hat{z}^i\right\}_{i=1}^{N}$
 9: **end for**

---

Note the simplicity of the IT-IN algorithm – it only involves exploration and supervised learning; there are no rewards, and the loss function is routine. In the following, we provide empirical evidence that, perhaps surprisingly – given the strict conditions for convergence of iterative inversion – IT-IN yields well-performing policies on several nontrivial tasks.

## 5 EXPERIMENTS

In this section, we evaluate IT-IN on several domains. Our investigation is aimed at studying the unique features of IT-IN and especially, the *steering* behavior that we expect to observe. We start by describing our evaluation domains, and implementation details that are common to all our experiments. We then present a series of experiments aimed at answering specific questions about IT-IN. To appreciate the learned behavior, we encourage the reader to view our supporting video results at: https://sites.google.com/view/iter-inver.

COMMON SETTINGS:

**VQ-VAE Intents:** For all our experiments, we generate intents using a VQ-VAE embedding of a rendered video of the trajectory. Rendering settings are provided next for each environment. We use VideoGPT's VQ-VAE implementation (Yan et al., 2021). An input video of size $64 \times 64 \times T$ (w, h, t) is encoded into a $16 \times 16 \times T/4$ integer intent $z^i$ given a codebook of size 50. Each integer represents a float vector of length 4. The training of the VQ-VAE is not the focus of this work, and we detail the training data for each VQ-VAE separately for each domain in the supplementary material. We remark that by visually inspecting the reconstruction quality, we found that our VQ-VAEs generalized well to the trajectories seen during learning.

**GPT-based policies and exploration noise** The policy architecture is adapted from VideoGPT (Yan et al., 2021), and consists of 8 layers, 4 heads and a hidden dimension of size 64. The model is conditioned on the intent via cross-attention. In the supplementary material, we report similar results with a GRU-based policy. Our exploration noise adds a Gaussian noise of scale $\eta$ to the action output.

**Evaluation Protocol**: While our algorithm only uses a loss on actions, a loss on the resulting trajectories is often easier to interpret for measuring performance. We measure the sum of Euclidean distances between agent state variables, accumulated over time, as a proxy for trajectory similarity; in our results, this measure is denoted as MSE. Except when explicitly noted otherwise, all our results are evaluated on test trajectories (and corresponding intents) that were not in the steering data, but were generated from the same trajectory distribution. None of the trajectories we plot or our video results are cherry picked.

DOMAINS

**2D Particle:** A particle robot is moved on a friction-less 2D plane, by applying a force $F = [F_X, F_Y]$ for a duration of $\Delta t$. The observation space includes the positions and velocities of the particle $S = [X, Y, V_X, V_Y]$, and motion videos are rendered using Matplotlib Animation (Hunter, 2007).

While relatively simple for control, this environment allows for distinct and diverse behaviors that are easy to visualize. In particular, we experiment with 2 behavior classes, for which we procedurally created training trajectories: (1) `Spline` motion, and (2) `Deceleration` motion. Both require highly coordinated actions, and are very different from the motion that a randomly initialized policy induces. Full details about the datasets are described in Appendix B.1.1.

**Reacher:** A 2-DoF robotic arm from OpenAI Gym's Mujoco Reacher-v2 environment (Brockman et al., 2016). While usually in Reacher-v2 the agent is rewarded for reaching a randomly generated target, the goal in our setting is for the policy to reconstruct the whole arm motion, as given by the intent, which is encoded from a video of the motion rendered using Mujoco (Todorov et al., 2012). We handcrafted a trajectory dataset, termed `FixedJoint`, which is fully described in Appendix B.2.1.

**Hopper:** From OpenAI Gym's Mujoco Hopper-v2 environment (Brockman et al., 2016). The dataset is from D4RL's `hopper-medium-v2` (Fu et al., 2020), and consists of mostly forward hopping behaviors. There are several challenges in this domain: (1) the dynamics are non-linear, and include a non-smooth contact with the ground; (2) the desired behavior (hopping) is very different from the behavior of an untrained policy (falling), and requires applying a very specific force exactly when making contact with the ground (a 'bottleneck' in the state space); and (3) the camera is fixed on the agent, and forward movement can only be inferred from the movement of the background.

STEERING EVALUATION

The first question we investigate is whether IT-IN indeed steers learning towards the desired behavior. To answer this, we consider domains where the desired behavior is *very different* from the behavior of the initial random policy – the `Spline` and `Deceleration` motions for the particle, and the hopping behavior for `Hopper-v2`. As we show in Figure 2 (for particle), and Figure 3 (for `Hopper-v2`), IT-IN produces a policy that can track the desired behavior with high accuracy. We further show, in Figure 5 and Figure 6 in the supplementary material, that IT-IN works well for different trajectory lengths $T$.

Another question is whether IT-IN really steers the policy towards the desired trajectories, or perhaps improves some general properties of the policy, allowing a generally better reconstruction. We explore this question by a *cross-evaluation* – evaluating the performance of a policy trained with steering intents from `Particle:Splines` on test intents from `Particle:Deceleration`, which we will refer to as out-of-distribution intents, and vice versa. Interestingly, as Table 1 shows, performance on out-of-distribution intents is significantly worse than the performance that would have been obtained by training the policy with these intents as the steering dataset, and is even worse or comparable to training with no steering at all (cf. Table 2).

We also evaluated the importance of the exploration noise. We tested `Splines` with $T = 64$ and `Hopper-v2` with $T = 128$ with and without exploration noise, and a large $D_{steer}$ (2180 for Hopper-v2, 500 for Particle). As the results in Table 6 in the supplementary material show, exploration noise $\eta$ is crucial for the training procedure to converge towards the desired behavior. Relating this observation to our theoretical analysis, we believe that exploration improves the conditioning of the supervised learning problem.

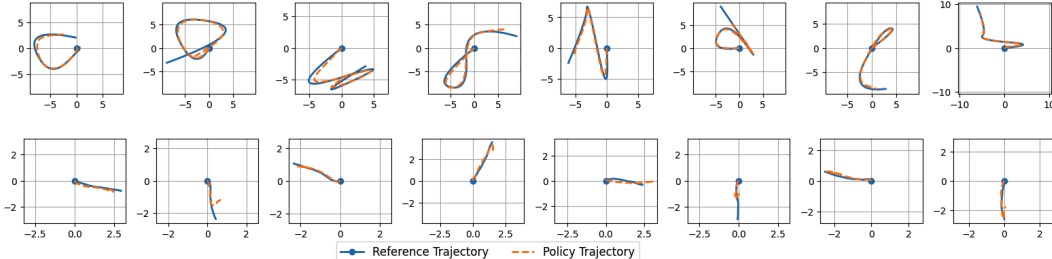

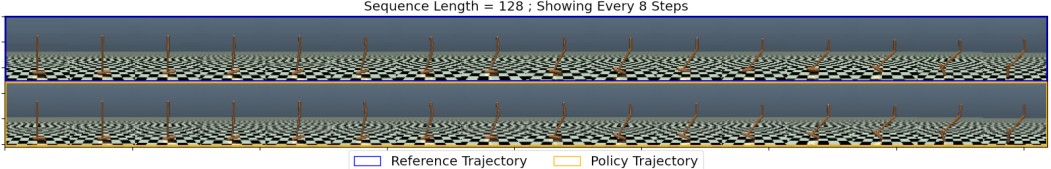

Figure 2: Particle results on `Splines` (top) and `Deceleration` (bottom). Here $T = 64$ and $|D_{\text{steer}}| = 500$. All trajectories start at `(0,0)`, marked by a blue circle. In `Deceleration`, the particle quickly decelerates to a stop at $t = 32$ – note the small overshoot at the end of each reconstructed trajectory, due to imperfect reconstruction of stopping in place.

Figure 3: Trajectory reconstructions in `Hopper-v2`, with $T = 128$ and $|D_{\text{steer}}| = 500$. Additional rollouts are presented in Appendix C.5 and in the supporting video results.

## STEERING DATASET SIZE AND GENERALIZATION

We next evaluate the generalization performance of IT-IN to intents that were not seen in the data, but correspond to state trajectories drawn from $P(\tau_s)$. To investigate this, we consider a domain where the desired behavior is *very diverse* – the `Spline` motions for the particle. We also report results on domains where the behavior is less diverse, such as `Hopper-v2` and `Deceleration` motions for particle. Naturally, we expect generalization to correlate with $M$, the size of $D_{\text{steer}}$. As our results in Table 2 show, additional steering data indeed improves generalization to unseen trajectories, albeit with diminishing returns as the amount of steering data is increased. As expected, in the more diverse distribution there was more gain to reap from additional data (significant improvement up to $|D_{\text{steer}}| = 50$), compared with the less diverse domain (most of the improvement is achieved already with $|D_{\text{steer}}| = 10$). Trajectory visualizations for `Splines` with different sizes of $D_{\text{steer}}$ are shown in Appendix C.2.

## COMPARISON WITH RL BASELINES

We compare IT-IN with reward-driven RL baselines. We consider the `Particle:Splines` environment, and two reward functions: (1) `STATE-MSE`: MSE between desired position and current position, and (2) `INTENT-MSE`: a sparse reward that is the MSE between the intents of the desired trajectory and the executed trajectory, given at the end of the episode. `STATE-MSE` is privileged compared to IT-IN and is arguably stronger than any IRL method in this task, as the reward is dense, and exactly captures the desired behavior. Any IRL method will run RL in an inner loop, with a reward that is less precise. `INTENT-MSE` is motivated by the fact that IT-IN effectively learns some similarity measure in intent space, and this reward captures this idea explicitly.

We used exactly the same policy architecture for all comparisons. We found that both RL methods did not train well with the GPT-based policy architecture[2], therefore we report results for the GRU

---

[2]Difficulty of RL with transformers was discussed in (Parisotto et al., 2020; Hausknecht & Wagener, 2022).

| | | Steering Dataset ($|D_{\text{steer}}| = 500$) | |
| | | Splines | Deceleration |
|---|---|---|---|
| Test Dataset | Splines | 69.2 | 210.9 |
| | Deceleration | 28.1 | 18.5 |

Table 1: Steering cross-evaluation. See Appendix C.8 for corresponding trajectory visualizations.

Table 2: Steering Dataset Size and Generalization. Here $T = 64$, and we show MSE averaged over 3 random seeds. Note that $|D_{\text{steer}}| = 0$ represents the case where no steering is used at all. In this case, we use trajectories sampled from a random policy to initialize $|D_{\text{prev}}|$ (see Algorithm 2). (*) For `Hopper-v2`, the maximal $|D_{\text{steer}}|$ is 1740 due to a limited amount of data in D4RL.

| | $|D_{\text{steer}}| = 0$ | $|D_{\text{steer}}| = 10$ | $|D_{\text{steer}}| = 50$ | $|D_{\text{steer}}| = 100$ | $|D_{\text{steer}}| = 500$ | $|D_{\text{steer}}| = 2000$ * |
|---|---|---|---|---|---|---|
| Particle:Splines | 199.7 | 105.4 | 75.8 | 72.7 | 69.2 | 66.9 |
| Particle:Deceleration | 30.0 | 20.5 | 21.1 | 20.2 | 17.9 | 18.6 |
| Hopper-v2 | 173.3 | 68.2 | 67.2 | 64.9 | 67.0 | 63.0 |

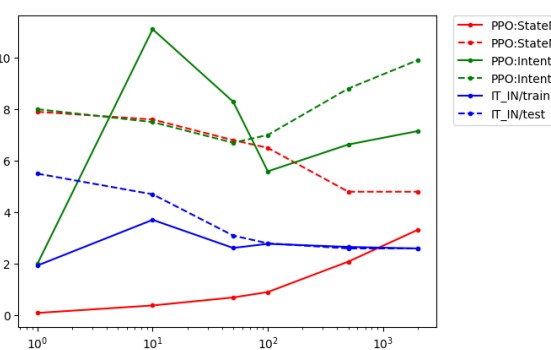

Figure 4: Comparison of IT-IN with RL Baselines. All results are MSE (lower is better) averaged over 3 random seeds. Note that IT-IN outperforms RL on test trajectories (dashed lines) for all $D_{\text{steer}}$ sizes.

policy, which is described in detail in the supplementary material. We used PPO (Schulman et al., 2017) for RL training, based on the implementation of Kostrikov (2018).

In Figure 4, we report results both on a held-out test set of trajectories, and on the training trajectories. As expected, for a small $D_{\text{steer}}$, STATE-MSE obtains near perfect reconstruction of training trajectories, yet high error on test trajectories, as the precise reward makes it easy for PPO to overfit. Interestingly, however, when increasing the size of $D_{\text{steer}}$, it becomes more difficult to overfit with PPO, even with the STATE-MSE reward. Note that for $|D_{\text{steer}}| = 2000$, the performance of STATE-MSE on training is worse than the performance of IT-IN on test! We are not aware of studies that investigated RL training of policies conditioned on very diverse contexts, and our results suggest that vanilla PPO is not well suited to this task. Importantly, on test data, IT-IN significantly outperforms both RL methods for all $D_{\text{steer}}$ sizes. We attribute this finding to the combination of stable supervised learning updates, and not relying on a reward. Finally, our results for INTENT-MSE do not come close to IT-IN, which we attribute to the more difficult learning from sparse reward.

## 6 DISCUSSION

We presented a new formulation for learning control, based on an inverse problem approach, and demonstrated its application to learning deep neural network policies that can reconstruct diverse behaviors, given an embedding of the desired trajectory. We developed the fundamental theory underlying iterative inversion, and demonstrated promising results on several simple tasks.

We only considered a particular trajectory embedding based on an off-the-shelf VQ-VAE, which we believe to be general and practical. Important questions for future work include characterizing the effect of the embedding on performance, and training an embedding jointly with the policy. Additionally, the exploration noise, which we found to be important, can potentially be replaced with more advanced exploration strategies from the RL literature.

Another interesting question is how to generate intents from a partial description of a trajectory, such as a natural language description. Diffusion models, which have recently gained popularity for learning distributions over latent variables (Rombach et al., 2021), are one potential approach for this.

One open question that remains is the gap between the strict conditions for convergence under a linear approximation in our theory, and the generally stable performance we observed in practice with expressive policies and non-linear dynamics. Another open question is whether iterative inversion can be extended to non-deterministic systems. We believe that our work provides the fundamentals for further investigating these important questions.

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

# A PROOFS

## A.1 PROOF OF EQUATION 4

Throughout this and the rest of the theoretical proofs, with a slight abuse of notation, when a vector $u \in \mathbb{R}^N$ is added to a matrix $A \in \mathbb{R}^{M \times N}$, the addition is row-wise: $A + u \equiv A + \mathbf{1}u$ where $\mathbf{1}u = (u, \dots, u)^T \in \mathbb{R}^{M \times N}$.

Denote $Y = (y_1, \dots, y_M) \in \mathbb{R}^{M \times dim(\mathcal{Y})}$. The approximated linear function is $\mathcal{G}_{\Theta,b}(Y) = Y\Theta + b$ where $\Theta \in \mathbb{R}^{dim(\mathcal{Y}) \times dim(\mathcal{X})}$ and $b \in \mathbb{R}^{1 \times dim(\mathcal{X})}$ (note the added parameter $b$ to account for the bias). At iteration $n + 1$:

$$\Theta_{n+1}, b_{n+1} = \arg\min_{\Theta, b} \|\mathcal{F}(X^n)\Theta + b - X^n\|^2$$

This is an ordinary linear least squares problem with the solution

$$\Theta_{n+1} = (\mathcal{F}(X^n) - \overline{\mathcal{F}(X^n)})^{\dagger}(X^n - \overline{X^n}), \qquad b_{n+1} = \overline{X^n} - \overline{\mathcal{F}(X^n)}\Theta_{n+1} \qquad (6)$$

Then

$$X^{n+1} = \mathcal{G}_{\Theta_{n+1}, b_{n+1}}(Y) = Y\Theta_{n+1} + b_{n+1} = \overline{X^n} + (Y - \overline{\mathcal{F}(X^n)})\Theta_{n+1}$$

and averaging over the points obtains the result:

$$\overline{X^{n+1}} = \overline{X^n} + (\overline{Y} - \overline{\mathcal{F}(X^n)})\Theta_{n+1}$$

## A.2 PROOF OF THEOREM 1

Using the notation defined in AppendixA.1.
Assuming $\mathcal{F}(X) = XF + h$ is a linear function with $F \in \mathbb{R}^{dim(\mathcal{X}) \times dim(\mathcal{Y})}$ and $h \in \mathbb{R}^{1 \times dim(\mathcal{Y})}$.
Assuming $rank(\mathcal{F}(X^0) - \overline{\mathcal{F}(X^0)}) = dim(\mathcal{Y})$ then $(\mathcal{F}(X^0) - \overline{\mathcal{F}(X^0)})^T(\mathcal{F}(X^0) - \overline{\mathcal{F}(X^0)})$ is invertible and $\Theta_1$ defined on Equation 6 is well defined.
Then using the fact that $\mathcal{F}(X^n) - \overline{\mathcal{F}(X^n)} = (X^n - \overline{X^n})F$:

$$\Theta_1 = (\mathcal{F}(X^0) - \overline{\mathcal{F}(X^0)})^{\dagger}(X^0 - \overline{X^0}) = \left((X^0 - \overline{X^0})F\right)^{\dagger}(X^0 - \overline{X^0})$$

and it satisfies

$$\Theta_1 F = I$$

where $I$ is the identity matrix. The bias term, according to Equation 6, is

$$b_1 = \overline{X^0} - \overline{\mathcal{F}(X^n)}\Theta_1 = \overline{X^0} - (\overline{X^0}F + h)\Theta_1$$

At the end of iteration 1, $X^1 = Y\Theta_1 + b_1$ and its matching outputs equal to the desired outputs:

$$\begin{aligned}
\mathcal{F}(X^1) &= X^1 F + h \\
&= Y\Theta_1 F + b_1 F + h \\
&= Y + (\overline{X^0} - (\overline{X^0}F + h)\Theta_1)F + h \\
&= Y + (\overline{X^0}F - \overline{X^0}F - h) + h \\
&= Y
\end{aligned}$$

## A.3 PROOF OF THEOREM 2

For clarity in the representation, we will use the following notation: $J_n^{-1} \equiv J^{-1}(X^n)$, $J_n \equiv J(\overline{X^n})$, $\tilde{\mathcal{F}}_n \equiv \overline{\mathcal{F}(X^n)}$ and $\mathcal{F}_n \equiv \mathcal{F}(\overline{X^n})$. We also define $H_n \equiv \mathcal{F}_n - \overline{Y}$ and $\tilde{H}_n \equiv \tilde{\mathcal{F}}_n - \overline{Y}$

First we show that $\tilde{J}_n^{-1}$ is non-singular. Since $\delta < \frac{1}{\zeta\beta}$ then $\rho(J_n\Delta_n J_n^{-1}) \leq \|J_n\Delta_n J_n^{-1}\| \leq \delta\zeta\beta < 1$ where $\rho(A)$ denotes the spectral radius of $A$. Therefore $(I + J_n\Delta_n J_n^{-1})$ is non-singular and $\tilde{J}_n^{-1} = J_n^{-1}(I + J_n\Delta_n J_n^{-1})$ is non-singular as a multiplication of non-singular matrices.

We denote by $\tilde{J}_n \equiv \left( \tilde{J}_n^{-1} \right)^{-1}$ its inverted matrix, and obtain the following bounds:

$$\|H_n - \tilde{H}_n\| = \|\mathcal{F}_n - \tilde{\mathcal{F}}_n\| \leq \lambda \tag{7}$$

$$\|\tilde{J}_n^{-1}\| = \|(I + \Delta^n)J_n^{-1}\| \leq \|J_n^{-1}\|(1 + \|\Delta^n\|) \leq \beta(1 + \delta) \tag{8}$$

$$\|\tilde{J}_n - J_n\| \overset{(1)}{\leq} \frac{\|J_n\|^2\|\Delta_n J_n^{-1}\|}{1 - \|J_n \Delta_n J_n^{-1}\|} \leq \frac{\|J_n\|^2\|\Delta_n\|\|J_n^{-1}\|}{1 - \|J_n\|\|\Delta_n\|\|J_n^{-1}\|} \leq \frac{\zeta^2\delta\beta}{1 - \zeta\delta\beta} \equiv \mu \tag{9}$$

$$\|\tilde{J}_n\| \leq \|\tilde{J}_n - J_n\| + \|J_n\| \leq \mu + \zeta \tag{10}$$

$$\|\tilde{\mathcal{F}}_n - \overline{Y}\| = \|\tilde{H}_n\| = \|(\overline{X^{n+1}} - \overline{X^n})\tilde{J}_n\| \leq \|\tilde{J}_n\|\|\overline{X^{n+1}} - \overline{X^n}\| \leq (\mu + \zeta)\|\overline{X^{n+1}} - \overline{X^n}\| \tag{11}$$

Inequality $(1)$ is developed in Horn & Johnson (2012, p. 381). Also note that the rest of the inequalities in (9) are well defined since $\delta < 1/\zeta\beta$.

The proof now continues similarly to the proof of Ortega & Rheinboldt (2000, 12.3.3). We set $G_n = \overline{X^n} - \tilde{H}_n\tilde{J}_n^{-1} = \overline{X^{n+1}}$, and show that $G_n$ is an Iterated Contraction:

$$
\begin{aligned}
\|\overline{X^{n+2}} - \overline{X^{n+1}}\| = \|\overline{X^{n+1}} - \tilde{H}_{n+1}\tilde{J}_{n+1}^{-1} - \overline{X^{n+1}}\| &= \|\tilde{H}_{n+1}\tilde{J}_{n+1}^{-1}\| \overset{(2)}{\leq} \beta(1+\delta)\|\tilde{H}_{n+1}\| \\
&\leq \beta(1+\delta)\|\tilde{H}_{n+1} - \tilde{H}_n - (\overline{X^{n+1}} - \overline{X^n})\tilde{J}_n\| \\
&\overset{(3)}{\leq} \beta(1+\delta)\|\tilde{H}_{n+1} - \tilde{H}_n - (\overline{X^{n+1}} - \overline{X^n})J_n\| + \beta(1+\delta)\|\tilde{J}_n - J_n\|\|\overline{X^{n+1}} - \overline{X^n}\| \\
&\overset{(4)}{\leq} \beta(1+\delta)\left(2\lambda + \|H_{n+1} - H_n - (\overline{X^{n+1}} - \overline{X^n})J_n\|\right) + \beta(1+\delta)\mu\|\overline{X^{n+1}} - \overline{X^n}\| \\
&\overset{(5)}{\leq} \beta(1+\delta)\left(2\lambda + \gamma\|\overline{X^{n+1}} - \overline{X^n}\|\right) + \beta(1+\delta)\mu\|\overline{X^{n+1}} - \overline{X^n}\| \\
&\leq \beta(1+\delta)\left(\frac{2\lambda}{\|\overline{X^{n+1}} - \overline{X^n}\|} + \gamma + \mu\right)\|\overline{X^{n+1}} - \overline{X^n}\| \\
&\overset{(6)}{\leq} \beta(1+\delta)\left(\frac{2\lambda(\mu+\zeta)}{\|\tilde{\mathcal{F}}_n - \overline{Y}\|} + \gamma + \mu\right)\|\overline{X^{n+1}} - \overline{X^n}\| \\
&= g(\|\tilde{\mathcal{F}}_n - \overline{Y}\|)\|\overline{X^{n+1}} - \overline{X^n}\|
\end{aligned}
$$

where inequality $(2)$ holds because of Bound 8, $(3)$ is the triangle inequality, $(4)$ is due to the Bounds 7 and 9 and the triangle inequality. Inequality $(5)$ is proven in Ortega & Rheinboldt (2000, 3.2.5) and inequality $(6)$ is from Bound 11.

Assuming $\beta(1+\delta)(\gamma + \mu) < 1$:

$$g(\|\tilde{\mathcal{F}}_n - \overline{Y}\|) = 1 \iff \|\tilde{\mathcal{F}}_n - \overline{Y}\| = \frac{2\lambda\beta(1+\delta)(\mu+\zeta)}{1 - \beta(1+\delta)(\mu+\gamma)} \equiv \rho$$

$g \geq 1$ only when $\tilde{\mathcal{F}}_n$ is close to $\overline{Y}$. $g$ is strictly-decreasing function of $\|\tilde{\mathcal{F}}_n - \overline{Y}\|$, thus if $\|\tilde{\mathcal{F}}_n - \overline{Y}\| \geq \rho + \epsilon$ for some $\epsilon > 0$ then $g(\|\tilde{\mathcal{F}}_n - \overline{Y}\|) \leq \alpha < 1$ where $\alpha$ is independent of $\|\tilde{\mathcal{F}}_n - \overline{Y}\|$.

The convergence of $\{\overline{X^n}\}$ follows from Ortega & Rheinboldt (2000, 12.3.2), as long as $g \leq \alpha < 1$. Since the following holds

$$\|\tilde{\mathcal{F}}_n - \overline{Y}\| \overset{(7)}{\leq} (\mu + \zeta)\|\overline{X^{n+1}} - \overline{X^n}\| \leq \cdots \leq (\mu + \zeta)\alpha^n\|\overline{X^1} - \overline{X^0}\|$$

then for every $\epsilon > 0$ there exists $k < \infty$ such that $\alpha^k \leq \frac{\rho + \epsilon}{(\mu+\zeta)\|\overline{X^1} - \overline{X^0}\|}$ and the convergence is towards the ball $\|\tilde{\mathcal{F}}_n - \overline{Y}\| \leq \rho$. Inequality $(7)$ is from Bound 11. For uniqueness, see the end of the proof of Ortega & Rheinboldt (2000, 12.3.3).

A.4   CONVERGENCE RESULTS FOR 1-DIMENSIONAL $\mathcal{F}$

We restrict ourselves to the 1-dimensional case, where $\mathcal{X} = \mathcal{Y} = \mathbb{R}$, and assume the function $\mathcal{F}$ is strictly monotone and its maximum and minimum slopes are not too different, thus the function is "close to" linear. We then show convergence at a linear rate.

Let $\mathcal{S}^{\mathcal{F}}(x_1, x_2) \equiv (\mathcal{F}(x_1) - \mathcal{F}(x_2))/(x_1 - x_2)$ denote the slope of $\mathcal{F}$ between $x_1$ and $x_2$, and $\max |\mathcal{S}^{\mathcal{F}}| \equiv \max_{x_1, x_2 \in \mathcal{X}} |\mathcal{S}^{\mathcal{F}}(x_1, x_2)|$ denote the maximum absolute slope of $\mathcal{F}$ and similarly $\min |\mathcal{S}^{\mathcal{F}}| \equiv \min_{x_1, x_2 \in \mathcal{X}} |\mathcal{S}^{\mathcal{F}}(x_1, x_2)|$ the minimum absolute slope.

**Assumption 4.** *$\mathcal{F}$ is continuous and strictly monotone, and $\frac{\max |\mathcal{S}^{\mathcal{F}}|}{\min |\mathcal{S}^{\mathcal{F}}|} \leq 2 - \epsilon$ for some $0 < \epsilon \leq 1$.*

**Theorem 3.** *Assume $\mathcal{X} = \mathcal{Y} = \mathbb{R}$, that Assumption 4 holds, and that there are only two desired outputs $M = 2$. Then for any $i \in \{1, 2\}$ and any iteration $n$: $|\mathcal{F}(x_i^{n+1}) - y_i| \leq (1 - \epsilon)|\mathcal{F}(x_i^n) - y_i|$.*

When the number of desired outputs is greater than 2, then convergence for each output is generally not guaranteed.

**Theorem 4.** *Assume $\mathcal{X} = \mathcal{Y} = \mathbb{R}$, that Assumption 4 holds and that at iteration $n$, $\forall i \; x_i^n < \mathcal{F}^{-1}(\overline{Y})$ or $\forall i \; x_i^n > \mathcal{F}^{-1}(\overline{Y})$. Then $\left| \overline{X^{n+1}} - \mathcal{F}^{-1}(\overline{Y}) \right| \leq (1 - \epsilon) \left| \overline{X^n} - \mathcal{F}^{-1}(\overline{Y}) \right|$.*

Theorem 4 guarantees that after finite iterations, the output segment intersects with the desired output segment. Note that Theorems 3 and 4 do not require any kind of approximations as in Assumption 3 nor for $\mathcal{F}$ to be differentiable.

### A.4.1   PROOF OF THEOREM 3

Denote $S_{max}^{\mathcal{F}} \equiv \max_{x_1, x_2} \mathcal{S}^{\mathcal{F}}(x_1, x_2)$ and similarly $S_{min}^{\mathcal{F}} \equiv \min_{x_1, x_2} \mathcal{S}^{\mathcal{F}}(x_1, x_2)$.

Assuming $\mathcal{X} = \mathcal{Y} = \mathbb{R}$. Then the approximated linear function is $\mathcal{G}_{\theta, b}(y) = y\theta + b$ where $\theta, b \in \mathbb{R}$ are scalars.
At iteration $n + 1$ and for $i \in [1, M]$:

$$x_i^{n+1} = \mathcal{G}_{\theta_{n+1}, b_{n+1}}(y_i) = y_i \theta_{n+1} + b_{n+1} \tag{12}$$

$$\theta_{n+1}, b_{n+1} = \arg\min_{\theta, b} \sum_{i=1}^{M} (\theta \mathcal{F}(x_i^n) + b - x_i^n)^2$$

**Lemma 5.** *if $\mathcal{X} = \mathcal{Y} = \mathbb{R}$ then $\forall n$: $\frac{1}{S_{max}^{\mathcal{F}}} \leq \theta_{n+1} \leq \frac{1}{S_{min}^{\mathcal{F}}}$ if $\mathcal{F}$ is strictly increasing and $\frac{1}{S_{min}^{\mathcal{F}}} \leq \theta_{n+1} \leq \frac{1}{S_{max}^{\mathcal{F}}}$ if $\mathcal{F}$ is strictly decreasing.*

*Proof.* We will prove for strictly increasing $\mathcal{F}$. The proof for strictly decreasing $\mathcal{F}$ is symmetrical. W.L.O.G assuming that $X^n$ is sorted: $\forall i$: $x_i^n \leq x_{i+1}^n$. Let $k > i$ then:

$$\mathcal{F}(x_i^n) + \mathcal{S}_{min}^{\mathcal{F}}(x_k^n - x_i^n) \leq \mathcal{F}(x_k^n) \leq \mathcal{F}(x_i^n) + \mathcal{S}_{max}^{\mathcal{F}}(x_k^n - x_i^n)$$

$$\frac{1}{\mathcal{S}_{max}^{\mathcal{F}}}(\mathcal{F}(x_k^n) - \mathcal{F}(x_i^n)) \leq x_k^n - x_i^n \leq \frac{1}{\mathcal{S}_{min}^{\mathcal{F}}}(\mathcal{F}(x_k^n) - \mathcal{F}(x_i^n))$$

$$
\begin{aligned}
\theta_{n+1} &= \frac{\frac{1}{M} \sum_{i=1}^{M} (x_i^n - \overline{X^n}) \left( \mathcal{F}(x_i^n) - \overline{\mathcal{F}(X^n)} \right)}{\frac{1}{M} \sum_{i=1}^{M} \left( \mathcal{F}(x_i^n) - \overline{\mathcal{F}(X^n)} \right)^2} = \\
&= \frac{\frac{1}{M^2} \sum_{i=1}^{M-1} \sum_{k=i+1}^{M} (x_k^n - x_i^n)(\mathcal{F}(x_k^n) - \mathcal{F}(x_i^n))}{\frac{1}{M^2} \sum_{i=1}^{M-1} \sum_{k=i+1}^{M} (\mathcal{F}(x_k^n) - \mathcal{F}(x_i^n))^2} \\
&\leq \frac{\frac{1}{M^2} \sum_{i=1}^{M-1} \sum_{k=i+1}^{M} \frac{1}{\mathcal{S}_{min}^{\mathcal{F}}} (\mathcal{F}(x_k^n) - \mathcal{F}(x_i^n))^2}{\frac{1}{M^2} \sum_{i=1}^{M-1} \sum_{k=i+1}^{M} (\mathcal{F}(x_k^n) - \mathcal{F}(x_i^n))^2} = \frac{1}{\mathcal{S}_{min}^{\mathcal{F}}}
\end{aligned}
$$

and

$$\theta_{n+1} \geq \frac{\frac{1}{M^2}\sum_{i=1}^{M-1}\sum_{k=i+1}^{M}\frac{1}{\mathcal{S}_{max}^{\mathcal{F}}}\left(\mathcal{F}(x_k^n)-\mathcal{F}(x_i^n)\right)^2}{\frac{1}{M^2}\sum_{i=1}^{M-1}\sum_{k=i+1}^{M}\left(\mathcal{F}(x_k^n)-\mathcal{F}(x_i^n)\right)^2} = \frac{1}{\mathcal{S}_{max}^{\mathcal{F}}}$$

$\square$

When $M = 2$, the regression line passes exactly at the points $(\mathcal{F}(x_1^n), x_1^n)$ and $(\mathcal{F}(x_2^n), x_2^n)$, and $b_{n+1}$ also takes the following forms:

$$b_{n+1} = x_1^n - \theta_{n+1}\mathcal{F}(x_1^n) = x_2^n - \theta_{n+1}\mathcal{F}(x_2^n)$$

Then placing $b_{n+1}$ in Equation 12 we get for every $i \in [1,2]$:

$$x_i^{n+1} = x_i^n + \theta_{n+1}(y_i - \mathcal{F}(x_i^n))$$

Denote the slope of $\mathcal{F}$ between $x_i^{n+1}$ and $x_i^n$: $\mathcal{S}^{\mathcal{F}}(x_i^{n+1}, x_i^n) \equiv \frac{\mathcal{F}(x_i^{n+1})-\mathcal{F}(x_i^n)}{x_i^{n+1}-x_i^n} = \frac{\mathcal{F}(x_i^{n+1})-\mathcal{F}(x_i^n)}{\theta_{n+1}(y_i-\mathcal{F}(x_i^n))}$
Then the following equations hold:

$$\mathcal{F}(x_i^{n+1}) = \mathcal{F}(x_i^n) + \theta_{n+1}\mathcal{S}^{\mathcal{F}}(x_i^{n+1}, x_i^n)(y_i - f(x_i^n))$$

$$y_i - \mathcal{F}(x_i^{n+1}) = \left(1 - \theta_{n+1}\mathcal{S}^{\mathcal{F}}(x_i^{n+1}, x_i^n)\right)(y_i - \mathcal{F}(x_i^n)) \tag{13}$$

Using Lemma5 and since $\mathcal{F}$ is always increasing or always decreasing, then $\theta_{n+1}\mathcal{S}^{\mathcal{F}}(x_i^{n+1}, x_i^n)) > 0$ and

$$\frac{1}{2-\epsilon} \leq \frac{\min|\mathcal{S}^{\mathcal{F}}|}{\max|\mathcal{S}^{\mathcal{F}}|} \leq \theta_{n+1}\mathcal{S}^{\mathcal{F}}(x_i^{n+1}, x_i^n)) \leq \frac{\max|\mathcal{S}^{\mathcal{F}}|}{\min|\mathcal{S}^{\mathcal{F}}|} \leq 2-\epsilon$$

$$\left|1 - \theta_{n+1}\mathcal{S}^{\mathcal{F}}(x_i^{n+1}, x_i^n))\right| \leq \max\left\{|1 - \frac{1}{2-\epsilon}|, |1-\epsilon|\right\} = 1-\epsilon \tag{14}$$

Then, placing it into Equation 13

$$\left|y_i - \mathcal{F}(x_i^{n+1})\right| = \left|1 - \theta_{n+1}\mathcal{S}^{\mathcal{F}}(x_i^{n+1}, x_i^n)\right||y_i - \mathcal{F}(x_i^n)| \leq (1-\epsilon)|y_i - \mathcal{F}(x_i^n)|$$

Note the convergence in one iteration for the linear case when $\epsilon = 1$.

### A.4.2    PROOF OF THEOREM 4

Denote $L_n$:

$$L_n \equiv \frac{\overline{Y} - \overline{\mathcal{F}(X^n)}}{\mathcal{F}^{-1}(\overline{Y}) - \overline{X^n}} = \frac{\frac{1}{M}\sum_{i=1}^{N}\overline{Y} - f(x_i^n)}{\frac{1}{M}\sum_{k=1}^{N}\mathcal{F}^{-1}(\overline{Y}) - x_k^n} = \sum_{i=1}^{M}\left(\frac{\mathcal{F}^{-1}(\overline{Y}) - x_i^n}{\sum_{k=1}^{M}\mathcal{F}^{-1}(\overline{Y}) - x_k^n}\right)\frac{\overline{Y} - f(x_i^n)}{\mathcal{F}^{-1}(\overline{Y}) - x_i^n}$$

$$= \sum_{i=1}^{M} w_{n,i}\frac{\overline{Y} - f(x_i^n)}{\mathcal{F}^{-1}(\overline{Y}) - x_i^n} = \sum_{i=1}^{M} w_{n,i}\,\mathcal{S}^{\mathcal{F}}(\mathcal{F}^{-1}(\overline{Y}),\, x_i^n)$$

Where $w_{n,i} = \frac{\mathcal{F}^{-1}(\overline{Y})-x_i^n}{\sum_{k=1}^{M}\mathcal{F}^{-1}(\overline{Y})-x_k^n}$, $\sum_{i=1}^{M} w_{n,i} = 1$ and, since we assumed $\forall i\ x_i^n < \mathcal{F}^{-1}(\overline{Y})$ or that $\forall i\ x_i^n > \mathcal{F}^{-1}(\overline{Y})$, then $\forall i\ w_{n,i} > 0$. Therefore $L_n$ is a weighted-mean of the slopes and $\mathcal{S}_{min}^{\mathcal{F}} \leq L_n \leq \mathcal{S}_{max}^{\mathcal{F}}$.
From Equation 4 the following holds:

$$\overline{X^{n+1}} - \overline{X^n} = \theta_{n+1}(\overline{Y} - \overline{\mathcal{F}(X^n)}) = \theta_{n+1}L_j(\mathcal{F}^{-1}(\overline{Y}) - \overline{X^n})$$

$$\mathcal{F}^{-1}(\overline{Y}) - \overline{X^{n+1}} = (1 - \theta_{n+1}L_j)(\mathcal{F}^{-1}(\overline{Y}) - \overline{X^n}) \tag{15}$$

Using Lemma 5 and the inequalities $\mathcal{S}_{min}^{\mathcal{F}} \leq L_n \leq \mathcal{S}_{max}^{\mathcal{F}}$, Inequality (14) from Appendix A.4.1 also applies for $L_n$, and we obtain:

$$|1 - \theta_{n+1}L_n| \leq 1 - \epsilon$$

$$\left|\mathcal{F}^{-1}(\overline{Y}) - \overline{X^{n+1}}\right| \leq (1-\epsilon)\left|\mathcal{F}^{-1}(\overline{Y}) - \overline{X^n}\right|$$

# B EXPERIMENTAL DETAILS

Table 3 contains a list of common hyperparameter values that we have used for all the experiments. Table 4 contains `Particle` and `Reacher-v2` specific hyperparameters, while Table 5 is listing `Hopper-v2` specific hyperparameters. We note that the minor difference in hyperparameter values between the domains evaluated is purposed only at achieving slightly better MSE results per domain. We observed that the steering behavior was relatively robust to hyperparameter values.

## B.1 PARTICLE ROBOT

The 2D plane in which the robot is allowed to move is finite, with the maximum coordinates increasing for longer horizons. When rendering the videos we include the entire 2D plane, up to the maximum coordinates. When evaluating policies, a validation set of 2,000 trajectories was used, which were unseen during training of the policies.

### B.1.1 DATASETS

**Splines** Trajectories follow the function of a B-spline curve[3]. The curves are of degree 2 with 5 control points, which are uniformly sampled between 0-1 in a 2-dimensional space.

**Deceleration** Random $X$ and $Y$ forces for the first $t_{acc}$ trajectory steps, and then $T - t_{acc}$ steps of deceleration, where $T$ is the time horizon. Deceleration at step $j > t_{acc}$ is done by setting $F_x^j = -\frac{1}{2}\frac{V_x^{j-1}}{\Delta t}, F_y^j = -\frac{1}{2}\frac{V_y^{j-1}}{\Delta t}$ (assuming the mass of the particle is 1). $V$ and $\Delta t$ are defined in Section 5. $\Delta_t = 0.1$

## B.2 REACHER-V2

### B.2.1 DATASETS

**Fixed Joint** Trajectories were collected to represent a scenario where one of the two robot arm joints is malfunctioning and is force fixed in place. The policy can only control the other robotic arm joint. When evaluating policies, a validation set of 2,000 trajectories was used, which were unseen during training of the policies.

## B.3 HOPPER-V2

### B.3.1 DATASETS

**Hopping** The datasets of size 2180 trajectories used for sequence-lengths 64 and 128 were extracted from D4RL's `hopper-medium-v2`, and consist of mostly forward hopping behaviors. When evaluating policies, a validation set of 436 trajectories was used, which were unseen during training of the policies. Unlike in the other evaluated domains, where trajectories sampled from a random policy were used to train the VQ-VAE, in `Hopper-v2` we have used input videos from D4RL's `Hopper-medium-v2` - the reason is that using the initial random policy, the trajectories terminated (hopper fell down) before reaching the desired $T$. For IT-IN training, we have modified `Hopper-v2` slightly so that the episode will not terminate when the Hopper falls, thus allowing it to reach T steps.

## B.4 GPT-BASED ARCHITECTURE

The model is conditioned on the intent via cross-attention. The actor network is comprised of 2 Linear layers of size 64, with a tanh activation.

## B.5 GRU-BASED ARCHITECTURE

The single-layer GRU's hidden state size is set to match the flattened intent size of 4096. The actor network is comprised of 2 hidden Linear layers of size 4096 and a tanh activation is used.

---

[3]https://en.wikipedia.org/wiki/B-spline

Table 3: Common hyperparameters for all experiments.

| Hyperparameter | Value |
|---|---|
| Learning rate | 5e-4 |
| Sampled rollouts per epoch ($N$) | 200 x Minibatch size |
| Training iterations | 2,000 |
| Training buffer size [rollouts] ($K$ x $N$) | 40 x N |
| Steering buffer $D_{\text{steer}}$ size [rollouts] | 500 |
| Ratio of steering intents in minibatch ($\alpha$) | 0.3 |
| Gradient norm clipping | 0.5 |
| GPT: # layers | 8 |
| GPT: # heads | 4 |
| GPT: hidden layer size | 64 |
| GPT: dropout | 0.2 |
| GPT: attention dropout | 0.3 |

Table 4: Particle & Reacher-v2 hyperparameters.

| Hyperparameter | Value |
|---|---|
| Minibatch size (rollouts) | 8 |
| Noise scale ($\eta$) | 4.0 |
| Total epochs ($n$) | 160 |

## C ADDITIONAL EXPERIMENTAL RESULTS

### C.1 PARTICLE:SPLINES - EFFECT OF TRAJECTORY LENGTH

We tested IT-IN on multiple horizons $T$ in the Splines domain, and found it to work well across horizons of 32, 64 and 128. We present sample visualizations with different $T$ values in Figure 5 (showing the final reconstructed trajectories) and in Figure 6 (showing trajectory progression during an epsiode).

### C.2 PARTICLE:SPLINES - EFFECT OF STEERING DATASET SIZE

In Figure 7 we present trajectory visualizations showcasing the effect of the size of the steering buffer $D_{\text{steer}}$ (cf. Table 2).

### C.3 PARTICLE:DECELERATION - EFFECT OF STEERING DATASET SIZE

Similarly to Section C.2, in Figure 8, we showcase the effect of the size of the steering buffer $D_{\text{steer}}$ (cf. Table 2) in the Particle:Deceleration domain.

### C.4 REACHER-V2

We present sample reconstruction visualizations for random-action trajectories from Reacher-v2 on 16-step sequences in Figure 10. Sample videos for 64-step FixedJoint sequences (trained with a GPT-based policy) can be found in the project's website: https://sites.google.com/view/iter-inver.

Table 5: Mujoco Hopper-v2 hyperparameters.

| Hyperparameter | Value |
|---|---|
| Minibatch size (rollouts) | 6 |
| Noise scale ($\eta$) | 1.0 |
| Total epochs ($n$) | 130 |

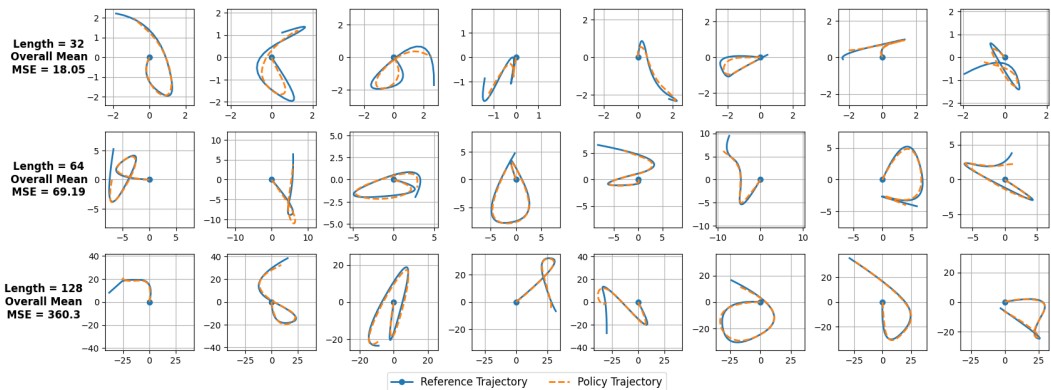

Figure 5: Example results on the `Splines` dataset, for different sequence lengths. In all cases shown here $|D_{\text{steer}}| = 500$. To the left of each row we state the average MSE on an evaluation set of 3 policies trained with different seeds. Note the increasing scale of the plots as the sequence length increases. Also note that all trajectories start at `(0,0)`, marked by the blue circle in each plot.

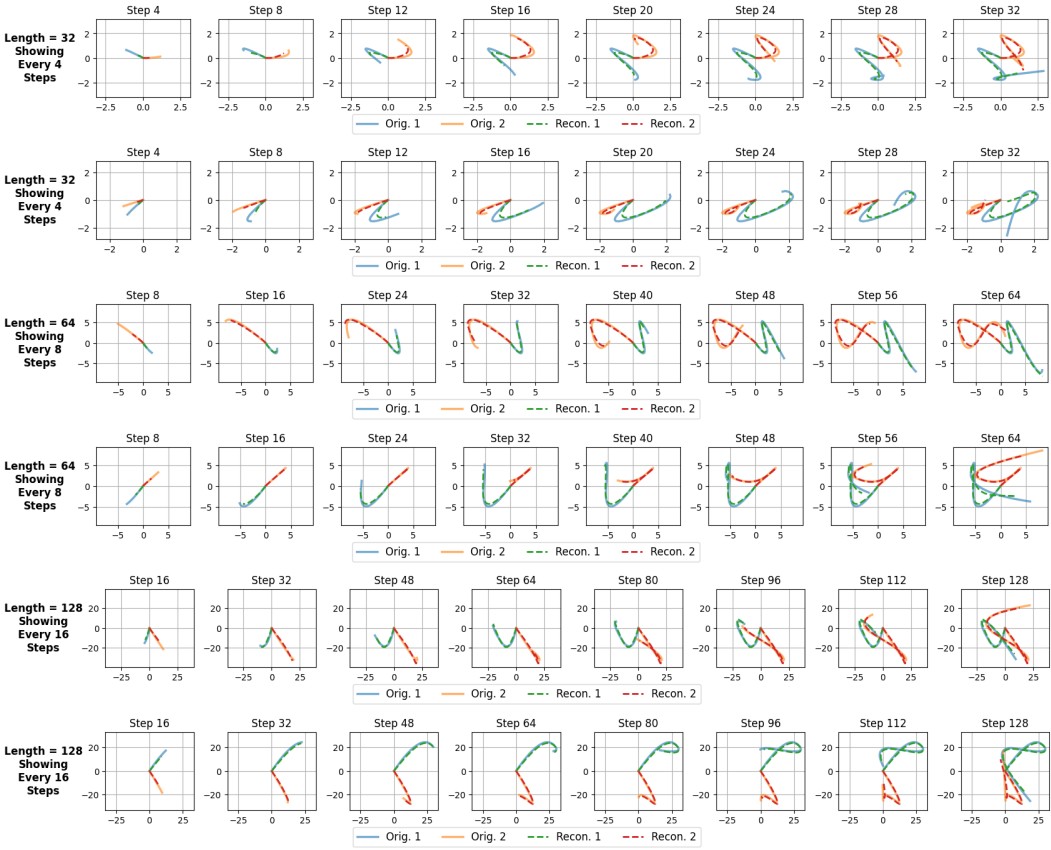

Figure 6: Visualization of trajectories progression in the `Splines` domain for different horizons $T$. Note the increasing scale of the plots as the sequence length increases. $|D_{\text{steer}}| = 500$.

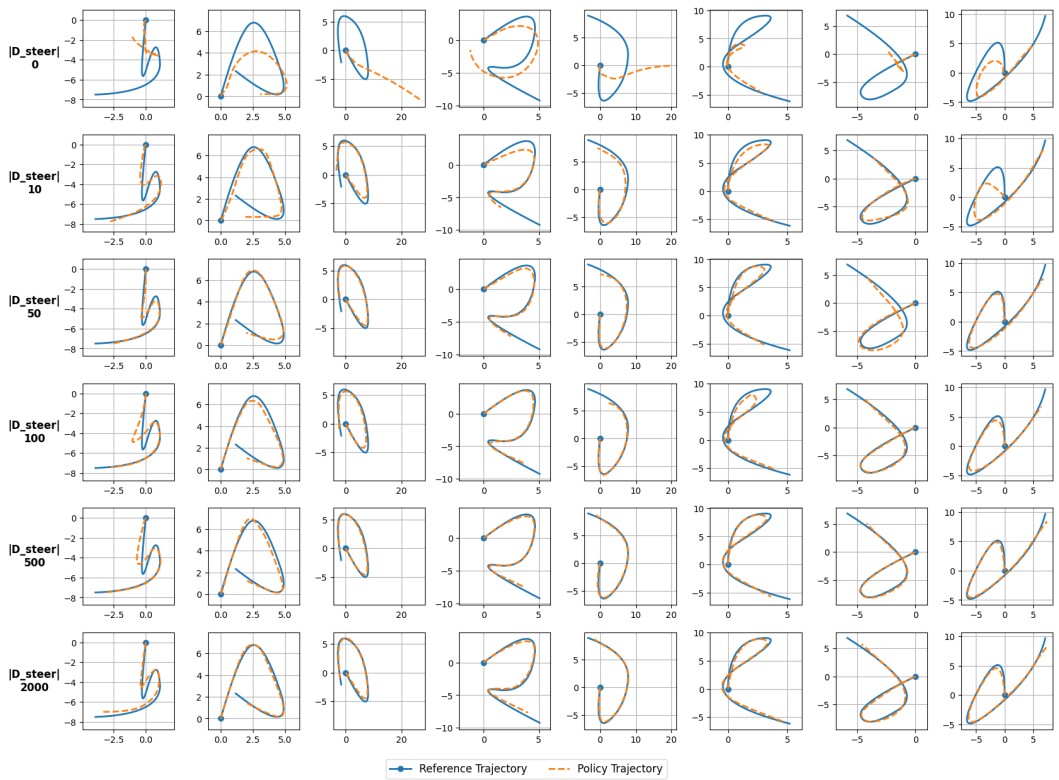

Figure 7: Example results on the `Particle:Splines` dataset for policies trained with different sizes of $D_{\text{steer}}$. Each row corresponds to a different size. Each column corresponds to a specific reference trajectory from the dataset, the intent of which was used as input to the policies. $T = 64$ was used in all experiments.

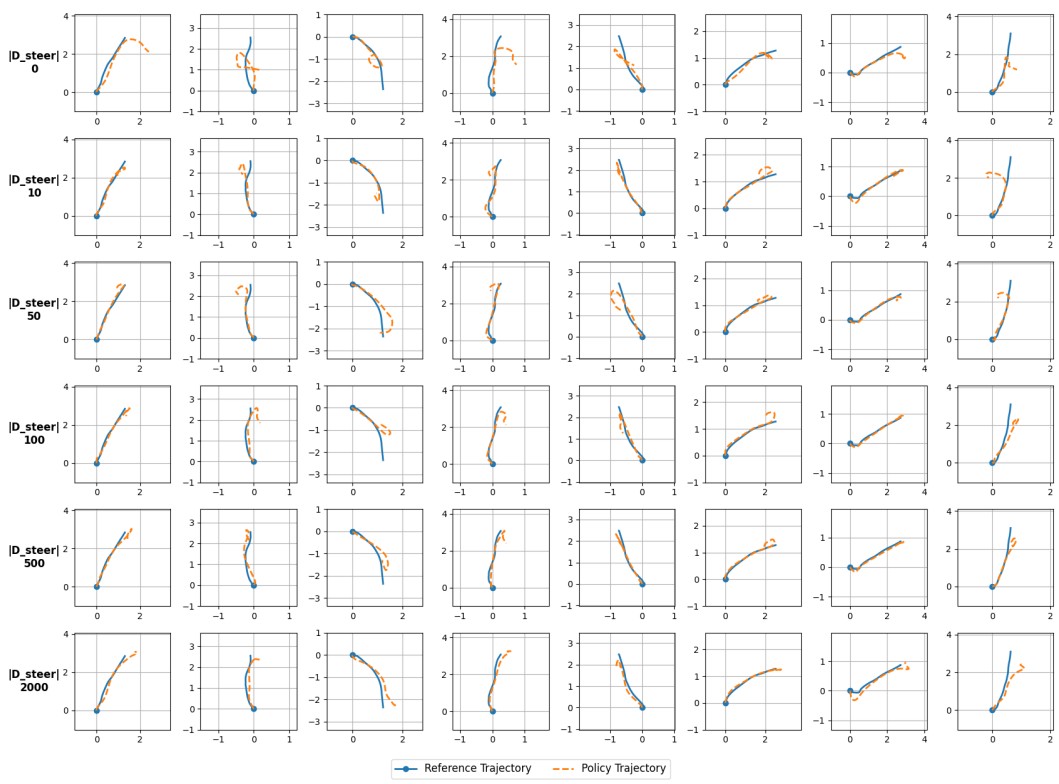

Figure 8: Example results on the `Particle:Deceleration` dataset for policies trained with different sizes of $D_{\text{steer}}$. $T = 64$ was used in all experiments. Figure structure same as in Figure 7.

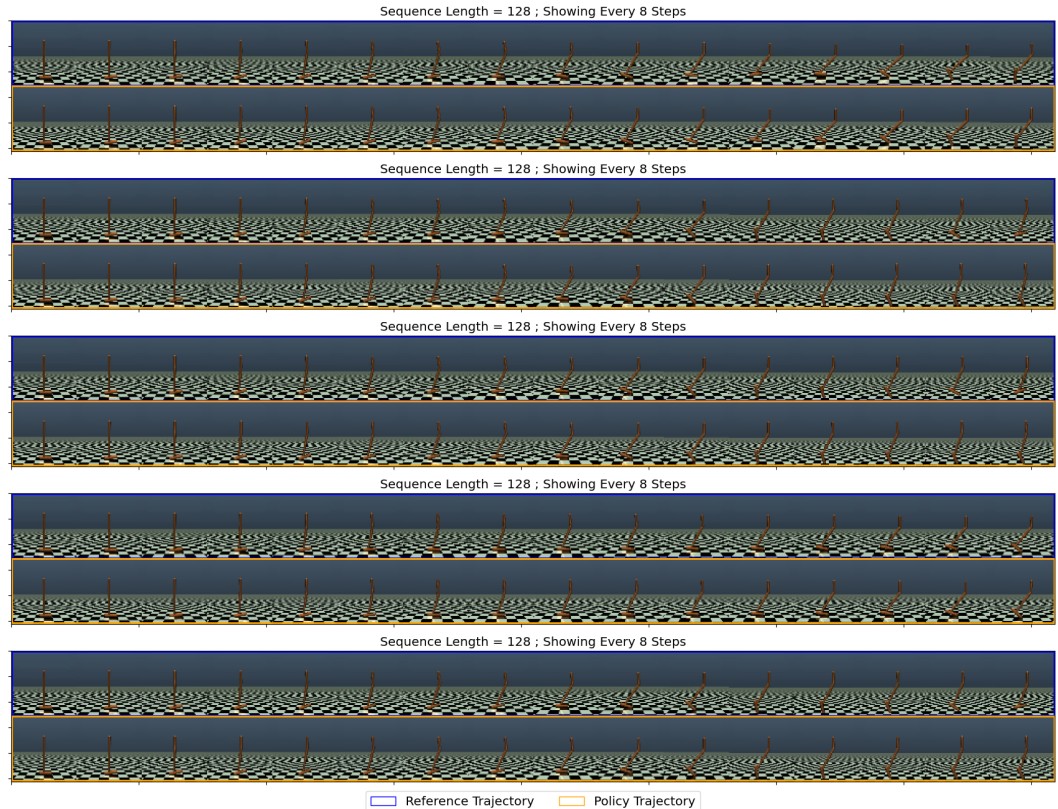

Figure 9: Examples of trajectory reconstructions in the `Hopper-v2` domain, with $T = 128$ and $|D_{\text{steer}}| = 500$.

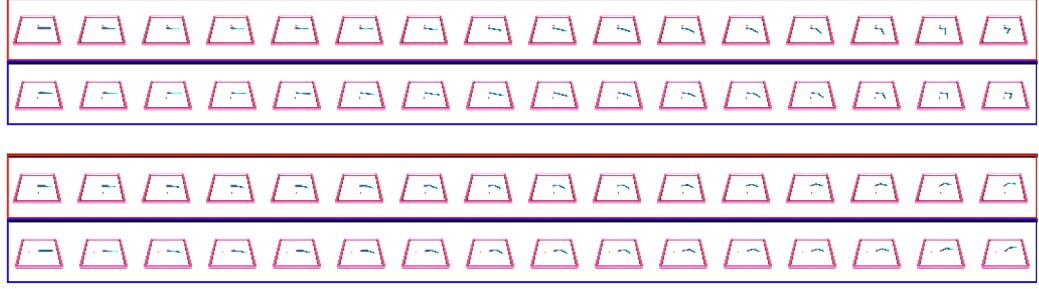

Figure 10: Examples of trajectory reconstructions in the Reacher-v2 domain. In each plot, the red row is the reference trajectory and the blue row is the policy reconstruction. These are based on a GRU policy. For ease of viewing, we modified the dark colors of the original rendered images.

## C.5  HOPPER-V2

We show additional examples of rollouts for the `Hopper-v2` domain on 128-step sequences in Figure 9.

## C.6  EXPLORATION

Table 6 is showcasing `Splines` and `Hopper-v2` reconstruction MSEs when trained with and without exploration noise $\eta$.

Table 6: Evaluation of policies trained with and without exploration. We show average MSE for 3 policies; due to different domains, MSEs are comparable only within each row.

| | Exploration Noise $\eta$ | No Exploration |
|---|---|---|
| `Particle:Splines`, $T = 64$, $|D_{\text{steer}}| = 500$ | 69.2 | 454.7 |
| `Hopper-v2`, $T = 128$, $|D_{\text{steer}}| = 1740$ | 483.3 | 920.6 |

Table 7: RL hyperparameters

| Hyperparameter | Value |
|---|---|
| PPO Clip Ratio | 0.2 |
| GAE $\lambda$ | 0.95 |
| Discount rate $\gamma$ | 0.99 |
| Learning rate | 1e-4 |
| Value loss coefficient | 0.5 |
| # epochs | 4 |
| # rollouts sampled per policy update | 128 |
| Total iterations | 5000 |

## C.7 RL BASELINE COMPARISON

Table 7 is summarizing the hyperparameters used for training RL policies with PPO Schulman et al. (2017), and a GRU policy.

## C.8 STEERING CROSS EVALUATION

In Figure 11 we show example rollouts from the experiments described in Section 5.

## C.9 GRU-BASED POLICY EXPERIMENTS

We report similar results with a GRU-based policy to the the results shown in Table 2 (analyzing the effect of steering dataset size) in Table 8, and similar results to Table 1 (steering cross-evaluation) in Table 9.

Table 8: Evaluation of IT-IN with a GRU policy on variable Steering Dataset size. $T = 16$. Note that $|D_{\text{steer}}| = 0$ represents the case where no steering is used at all. In this case, we use trajectories sampled from a random policy to initialize $|D_{\text{prev}}|$ (see Algorithm 2)

| | $|D_{\text{steer}}| = 0$ | $|D_{\text{steer}}| = 10$ | $|D_{\text{steer}}| = 50$ | $|D_{\text{steer}}| = 100$ | $|D_{\text{steer}}| = 500$ | $|D_{\text{steer}}| = 5000$ |
|---|---|---|---|---|---|---|
| `Particle:Splines` | 5.48 | 5.18 | 3.86 | 3.61 | 3.02 | 2.89 |
| `Particle:Deceleration` | 0.85 | 0.89 | 0.75 | 0.73 | 0.67 | 0.71 |
| `Reacher-v2:FixedJoint` | 2.49 | 2.05 | 1.68 | 1.64 | 1.58 | 1.61 |

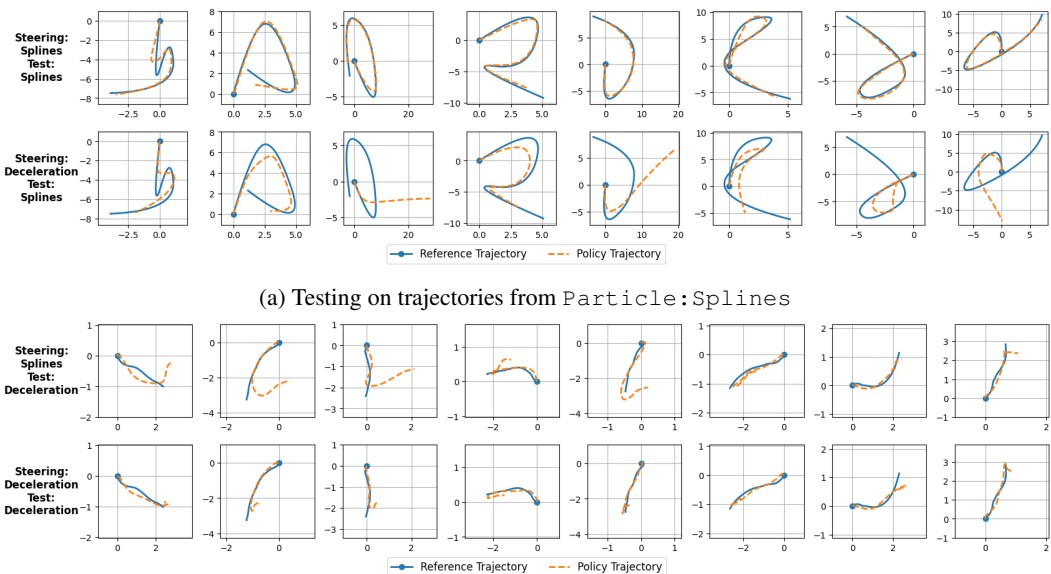

(a) Testing on trajectories from `Particle:Splines`

(b) Testing on trajectories from `Particle:Deceleration`

Figure 11: Examples comparing how policies trained with steering intents from either `Particle:Splines` or `Particle:Deceleration` perform when tested on trajectories from either datasets. We can see that when a policy is trained with steering intents from one dataset, it performs well on that dataset and performs poorly on the other. In each column the reference trajectory is the same.

Table 9: Steering cross-evaluation for a GRU-policy. $T = 16$.

|  |  | Steering Dataset ($|D_{\text{steer}}| = 500$) | |
|  |  | Splines | Deceleration |
| --- | --- | --- | --- |
| Test Dataset | Splines | 2.79 | 5.4 |
|  | Deceleration | 1.46 | 0.72 |