# OpenReview forum: "Learning Control by Iterative Inversion"
_ICLR.cc/2023/Conference — Submitted to ICLR 2023_

### Official Review · Reviewer_VhZe · 2022-10-24

**Confidence:** 3
**Correctness:** 4
**Technical Novelty And Significance:** 3
**Empirical Novelty And Significance:** 3
**Recommendation:** 6

**Clarity, Quality, Novelty And Reproducibility:**

The proposed method in this paper is novel. The idea makes sense to me. This paper is well written and proofs are easy to follow.

**Strength And Weaknesses:**

This paper presents a new viewpoint of the problem of learning control which directly mapping state sequence to action sequence.
Compared to the RL methods, the proposed algorithm does not need a pre-defined reward function which could be hard for many real applications. The analysis looks valid to me. And enough experiments are presented to support the statement in this paper.

The assumptions raised in this paper are pretty strict to satisfy in real applications. Like stated in section 3, the conditions can be interpreted as F can be well approximated by a linear function. This means the mapping from state sequence to the action sequence should be close enough to a linear function in order to have convergence guarantee.  Do these assumption hold in the examples under the experiment section?

I would also question the size of the steering dataset for more complex tasks with more intents. In order to learn the mapping function, it is necessary  to have the steering dataset sufficient enough to capture all the sequences needed for accomplishing tasks. Could you give more explanation about how to construct such dataset?

**Summary Of The Paper:**

This paper solves the problem of  learning for control by learning a inverse mapping between action sequence and states sequence.  To handle the distribution shift, the authors proposed  a general algorithm called interactive inversion which interactively  learn the inverse mapping under the current policy, then apply it to the desired output to obtain a new policy for it. Under certain assumptions, the proposed algorithm is proved to converge to the desired mapping theoretically.  By applying the iterative inversion, the authors further present a IT-IN algorithm for learning actions given any desired state sequence. Through experiments, the authors show the proposed algorithm has an improved performance on imitating diverse behaviors compared to reward based RL methods.

**Summary Of The Review:**

The proposed method is shown to perform significantly better than other works. Sufficient proofs are provided to support the statement in this paper.

---

> ### Author Response · Authors · 2022-11-09
> **Response**
>
> Thank you for your feedback!
>
> ## Assumptions in theory and in practice
>
> We agree - the theoretical assumptions are rather strict, and we find the fact that the method works robustly in practice rather surprising (this is an important part of our novelty!).
>
> We do not know how to verify whether the assumptions hold in our experiments. What we do know is that the dynamics in some experiments (e.g., Hopper) are far from linear. We do not know how to characterize the mapping from state sequence to video, and subsequently, from video to the VQ-VAE intent. We believe it is also far from linear.
>
> ## Size of steering dataset
> We are not sure we understand the reviewer’s question. In Table 2 we study the effect of the steering dataset size, and evidently, even small datasets can lead to substantial gains. If this does not answer your question, we would appreciate a clarification.

---

### Official Review · Reviewer_2R3c · 2022-10-26

**Confidence:** 4
**Clarity, Quality, Novelty And Reproducibility:** Paper is written clearly. Novelty is …
**Correctness:** 3
**Technical Novelty And Significance:** 2
**Empirical Novelty And Significance:** 2
**Recommendation:** 6

**Strength And Weaknesses:**

*Strengths*

- This is an interesting approach/problem. Learning to match a video demonstration through online RL is indeed an important and well studied problem, and the paper presents an interesting take on it.

*Weaknesses*

- The first major limitation of the work is the novelty of the proposed method. There is a vast literature of works that learn inverse dynamics models for learning to generate actions from observations, from Schaal et al. to more recent works like Ghosh et al. and Zheng et al. So fundamentally the idea of supervised learning of an inverse model to match demonstrations is not new, nor is the idea of using such an inverse model iteratively online during agent interaction (while GCSL trains a goal conditioned policy, extending the policy to follow the sequence of goals in a demonstration is a trivial extension). The main thing that appears to be new about the proposed approach is the method of training the inverse model, specifically through a VQ-VAE and a transformer that maps the full sequence of states to a sequence of actions directly (rather than operating on single s_t, s_t+1 pairs). I actually think this part of the contribution is quite interesting and novel, and could make sense as an improvement over prior works. I would advise the authors to emphasize this as the main contribution of the work.

- The second issue is with the experiments. While the toy experiment showing how the method works is nice, there really isn't a comprehensive comparison to the relevant baselines. Even the RL baselines used are only tangentially relevant to the proposed method. Fundamentally the proposed method is an inverse RL method (demonstrations --> online interaction to produce a policy that matches the demos). Therefore the work should compare to the state-of-the-art inverse RL methods like GAIL from observation or the various extensions like Berseth et al. and Reddy et al, as well as methods like GCSL (Ghosh et al.) on more challenging benchmarks like vision-based robotic manipulation.

Berseth et al. Towards Learning to Imitate from a Single Video Demonstration. 2019.
Reddy et al. SQIL: Imitation Learning via Reinforcement Learning with Sparse Rewards. ICLR 2020.
Schaal et al. Real-Time Robot Learning With Locally Weighted Statistical Learning. ICRA 2000.
Ghosh et al. Learning to Reach Goals via Iterated Supervised Learning. ICLR 2021.
Zheng et al. Semi-Supervised Offline Reinforcement Learning with Action-Free Trajectories. 2022.

**Summary Of The Paper:**

The paper presents iterative inversion, an approach for learning control from video demonstrations (without actions) and online interaction. The approach works as follows:
- Start with a random exploration policy
- Train an inverse dynamics model on the online data mapping a sequence of states to a sequence of actions
- Apply the inverse dynamics model to the demo trajectory to get the reference action sequence
- Update the policy to match the reference action sequence
- Repeat while the policy improves+explores until it matches the reference video demo.

There are a few additional tricks to stabilize thing but that is the essence of the method. The inverse dynamics modeling is done in a seq-to-seq fashion using a VQ-VAE image encoder and a transformer sequence model.

In experiments, they first confirm that the method indeed steers behavior toward the demos effectively in some toy tasks, then compares to some RL methods on image-based Hopper/Reacher.


**Summary Of The Review:**

Overall this is an interesting paper tackling an interesting problem. However, the work needs to better position its contribution w.r.t the many similar works in the field, and expand its experiments to more challenging domains and consider more relevant baselines.

--- Post rebuttal
The rebuttal has addressed most of my concerns.

---

> ### Author Response · Authors · 2022-11-07
> **Clarification of our contribution and novelty**
>
> We thank the reviewer for their feedback.
>
> We believe that the reviewer overlooked parts of our method, leading to a mis-evaluation of our contribution. The summary of our method should actually be:
> - Start with a random exploration policy
> - Train an inverse dynamics model on the online data mapping a sequence of states to a sequence of actions
> - ~~Apply the inverse dynamics model to the demo trajectory to get the reference action sequence~~
> - ~~Update the policy to match the reference action sequence~~
> - Use the demo trajectory embeddings as input to the policy and collect rollouts (with some exploration noise). **Compute embeddings for the rollout trajectories**
> - Fit the policy on the **collected** rollouts and **their embeddings**
> - Repeat while the policy improves+explores until it matches the reference video demo.
>
> **This difference may appear subtle, but is actually profound, and is the main idea we investigate in the paper**. That is - why should fitting to the *collected rollouts+their embedding* (data that no longer contains any signal from the demos) act to bring the policy closer to actions for the demos?
>
>
> Our contribution is formulating a mathematical framework to study this question, in Section 3 of our paper. This framework is novel, and very different from any prior work that we are aware of (the use of Newton’s method as the driving force behind the learning updates is very different from concepts such as contracting Bellman operators or policy gradient descent in RL, or distribution matching in IRL). Importantly - our analysis shows when such an iterative learning scheme could work, and can be the basis for future theoretical and algorithmic investigation of this problem.  We kindly ask the reviewer to take this contribution into account in their evaluation of our novelty.
>
> ## Inverse models literature
> The reviewer is absolutely correct, we should have discussed this in the related work, and we apologize. We are quite familiar with the literature on inverse dynamics models, and to our knowledge, all prior work collected data that is ‘in distribution’ with respect to the desired task. That is, if the hopper needs to hop, collect data relevant to hopping. Conventional methods cannot cope with a strong distribution shift - trying to hop when data is only collected from falling. The only exception we are aware of, which we learned about after the submission deadline, is the work of Hong et al. There, an RL agent was trained to steer data collection to areas where the inverse model errs. However, Hong et al. require the full state trajectory as input for the policy, require RL training as part of their method, and report difficulty in scaling to high dimensional action spaces, where the unsupervised curiosity-based exploration method cannot effectively steer towards relevant areas of the state space.
>
> ## References
>
> Hong et al. Adversarial active exploration for inverse dynamics model learning. CoRL 2019.

---

> > ### Author Response · Authors · 2022-11-07
> > **Clarification of our contribution and novelty (part 2)**
> >
> > ## Inverse RL
> > As we discuss in the related work section, while we solve a similar problem, our approach is very different, both theoretically and empirically. While more work is required to have a complete picture of which method is better for which tasks, we already provided one advantage - *not using an RL as a subroutine, but only supervised learning*.
> >
> > In our “Comparison with RL Baselines” section, we show that this allowed us to train GPT-based policies, which are currently difficult to train with RL.
> >
> > More importantly, to illustrate the advantage of not using RL, we focused on a setting where the desired trajectories are diverse, and the agent must reconstruct them from the intent. This is quite different from most IRL evaluations we are aware of, where there is no context variable such as the intent. For example, in the papers the reviewer mentions, all demos are from the same task, and the agent is evaluated on that task. While in principle the context can be regarded as an augmentation of the state, the fact that it is high dimensional (size 4096) and critical for inferring the task (what trajectory to follow), makes it a very different problem. As we show, on the simple particle domain, with the *dense MSE reward*, PPO is significantly outperformed by our method. We will add a comparison with GAIL, but there is little reason to believe that it will do better, as it builds on PPO/TRPO, and it's hard to imagine a better reward for this task than dense MSE. This should not come as a surprise - the direct backprop in supervised learning can more effectively identify patterns in the high-dimensional intent vector than the noisy policy gradient of RL.
> >
> > If the reviewer can point us to multi-task IRL works that can tackle high-dimensional contexts, we will be happy to add to our comparison. For conventional IRL methods, however, our current evaluation already clearly establishes they are not suited for this problem setting.
> >
> > ## GCSL
> > Indeed, our work is very related to GCSL, as we discussed in length in the related work section. Taking T=1 in our method and applying the policy iteratively along a reference trajectory will result in the “sequence of goals” variant of GCSL the reviewer suggested. On the other hand, taking the intent as the goal state exactly recovers the original GCSL.
> >
> > Our work can thus be seen as an extension of GCSL, where the embedding replaces the goal. This is more general than “sequence of goals”, as we do not assume the full reference trajectory as input, but only its embedding.
> >
> > Our contribution, however, is investigating the steering component of the method - an idea that originated in the GCSL paper (there, goals replace intents), but was not investigated at all to our knowledge. The theory in GCSL assumes no distribution shift (coverage of all goals in the data), and **the idea that steering can overcome distribution shift is, to the best of our knowledge, novel**. Our work therefore exposes this component of the GCSL method as one key to its success, and our paper extensively investigates this idea, empirically and theoretically. We believe our results strongly complement the current knowledge about GCSL-based methods.

---

> > > ### Author Response · Authors · 2022-11-15
> > > **Follow-up**
> > >
> > > Dear Reviewer, we would like to ask if your concerns over the novelty of our method and the related work have been addressed, or if there were any other issues that would prevent you from increasing your score. Please let us know, and thank you for your time and feedback.

---

> > > > ### Comment · Reviewer_2R3c · 2022-12-01
> > > > **Response**
> > > >
> > > > The rebuttal has addressed most of my concerns and I've raised my score.

---

### Official Review · Reviewer_9eeh · 2022-11-08

**Confidence:** 3
**Correctness:** 3
**Technical Novelty And Significance:** 2
**Empirical Novelty And Significance:** 2
**Recommendation:** 5

**Clarity, Quality, Novelty And Reproducibility:**

The exposition of relevant works may be enhanced by more direct comparison to imitation learning which seems to better match the problem setting of this work (for the absence of rewards). It should also be compared to other works on learning dynamics.

I also think that it might help motivate the proposed problem setting by providing some concrete motivating examples/application. If I am not mistaken, the experiments have reference trajectories whose actions we know but choose to forget making the experiments feel a little artificial. As you suggested with the intent/goal-conditioned experiments, how should one use the trained inverse model to control when we need to provide the next state?

Do you think that the assumption of bijectivity on $F$ is too restrictive for applying your approach? In particular, consider $T=1$ in (2), any different pairs of $(s_0, s_1)$ need to have different actions $a_0$.

**Strength And Weaknesses:**

Strengths:
1. The basic approach (algorithm) seems reasonable and novel. Instead of access to a loss measuring the similarity of state trajectories, it assumes access to a loss measuring the similarity of action sequences during training. However, it fails to explicitly discuss the connection and assumption between the two distances (over states and over actions). (see Weakness1)

Weaknesses:
1. The formal problem formulation and the exposition has gaps. If the ultimate goal is to reproduce the reference state trajectory (as suggested by the evaluation protocol), then we are assuming that low action-action loss implies low state-state loss, correct? I imagine that differences in early actions in a sequence would have a larger impact than the differences in later ones. How do MSE over states and over actions relate to each other in your experiments?
1. The proposed problem setting does not seem adequately motivated. Why do we want to learn an inverse model that is accurate over a small (in comparison to all possible trajectories) set of state trajectories? In learning dynamics via interactions, a key challenge is the difficulty to access many states (unlike in IID setting) with a simple policy, e.g., uniformly random policy. Are the reference trajectories helping us access such region of the state space?

**Summary Of The Paper:**

The authors try to solve the problem of inverting a deterministic dynamics model when query access to the said model (being able to query the next state given the current state and the current action). The performance of the trained inverse model is evaluated on a distribution of reference state trajectories.

The core idea proposed is to assume that the dynamics is invertible, and we try to find the inverse of the dynamics which we can apply to the observed state trajectory to find its actions. Specifically the mapping from a sequence of actions to a sequence of states $F : A^n \rightarrow S^{n-1}$ is invertible so $F^{-1}$ is well defined. Then we learn $F^{-1}$ with a parametric model $G_\theta$ such that $G_\theta \circ F \approx F^{-1} \circ F = \text{id}$. Crucially we assume access to a "reasonable" loss over the action sequences which we use to train $G_\theta$. The trajectories on which we optimize $G_\theta$ is a mixture of samples from the distribution of reference trajectories and trajectories from randomly interacting with $F$.

**Summary Of The Review:**

I like aspects of the proposed approach and the problem of learning inverse dynamics model. But the presentation could be made clearer. In particular, the problem should be formulated more clearly.

---

> ### Author Response · Authors · 2022-11-09
> **Response and a clarification of our contribution**
>
> Thank you for the feedback! We begin with clarifying our contribution, which we believe is not fully reflected in your evaluation.
>
> We contribute a **new learning principle**, based on the novel idea that steering can overcome distribution shift. This principle is very different from any prior work that we are aware of – the use of Newton’s method as the driving force behind the learning updates is very different from concepts such as contracting Bellman operators or policy gradient descent in RL, or distribution matching in IRL.
> **This discovery has value by itself - it extends our understanding of learning systems.** The fact that learning control does not require an RL component can appeal, for example, to Neuroscientists, which have been searching for reward correlates in the brain for decades.
>
> Our paper investigates this new principle in depth - we provide a non-trivial theory that explains its fundamental mechanism, and in the context of learning to track a trajectory from intent, we extensively evaluate the steering behavior, and the advantages that learning without RL provides.
>
> Our experiments focus on the somewhat narrow application of learning to track a trajectory from its intent, which has important applications (see below). However, often the technical advantages of a new learning principle take time to mature. A recent example of this is diffusion models (Sohl-Dickstein et al., Ho et al.).
>
> ## Clarification of our method:
> >The trajectories on which we optimize $G_\theta$ is a mixture of samples from the distribution of reference trajectories and trajectories from randomly interacting with $F$.
>
> This is not accurate; see also our response to Reviewer 2R3c. We only optimize $G_\theta$ on trajectories from interacting with $F$, and their intents. The *input to the policy* for generating these trajectories is the reference intent (mixed with intents from previous iteration). This subtle difference is important - why should fitting to the *collected rollouts+their intents* (data that no longer contains any signal from the reference) act to bring the policy closer to actions for the reference? We answer this question both theoretically and empirically.
>
> ## (1) Distances over states and over actions
> Our theory clearly addresses this: Assumption 2 requires bounded derivatives of $F$ and $F^{-1}$, which means that actions cannot produce unbounded changes in states, and vice versa.
>
> In our experiments, as we discuss at the top of Page 7, we measure MSE between trajectories because it is easier to interpret visually.
> We can add the MSE over actions to our results if the reviewer feels it is important - this can be done similarly to ‘teacher forcing’ - by feeding in the reference state trajectory as the history input to the policy and measuring the action at each time step.
>
> Impact of early action errors - the hopper domain is a good example of this - failing to bounce at the right time cannot be recovered from. Thus, our learned model is accurate enough to reconstruct trajectories, even though the impact of action errors can be large.
>
> ## (2) Motivation for reference trajectories
> Ideally, we would like our learned policy to be accurate over all possible trajectories. However, there are two reasons why this is difficult.
> - Exactly as the reviewer mentions, some trajectories are hard to reach by random exploration. The hopping behavior in our hopping experiment is exactly such a case.
> - The space of all possible trajectories is **huge**. In the 2D particle, for example, it is roughly exponential in T, and becomes impossible to cover during training. This domain, which may appear ‘toy’, is actually not trivial at all. In a preliminary investigation, we tried to generate training data for the inverse model using RL with various exploration methods (such as skew-fit and curiosity) only to obtain ‘wild’ trajectories that are very different from the desired distribution. This is somewhat in contrast to the success of various ‘goal-exploration’ methods, which only need to cover the reachable goals, but can ignore the trajectory towards the goal. Reference trajectories are necessary to focus on a space that can be large (e.g., see our splines dataset), but still feasible to learn.
>
> ## Imitation learning and inverse dynamics literature
> Please also see our response to Reviewer 2R3c. We are not aware of any imitation learning / IRL work that can handle multiple tasks specified by a high-dimensional context vector as in our problem setting. Also, conventional inverse dynamics models require the full state trajectory, and not an embedding, and cannot cope with strong distribution shifts.

---

> > ### Author Response · Authors · 2022-11-09
> > **Response (part 2)**
> >
> > ## Motivation
> > Our goal is to learn a policy that can ‘master’ a dynamical system, i.e., perform any desired trajectory in it. This can be beneficial for robotics, for example, a construction robot that can move earth to any desired form. Here, an intent can be produced from a simulation of the robot, for example, or in the future, from a generative model that takes as input a description of the desired trajectory.
> > Another application is labeling observation-only videos, as in the recent work on video pretraining (Baker et al.). They used human experts for collecting relevant data for the inverse model, and our work can potentially be used to relax this requirement.
> >
> > Note that our method does not specifically require the next state, but some embedding of the desired trajectory.
> >
> > ## Bijectivity
> > We assume the reviewer is worried that different actions cannot produce the same state change. In theory, it would require rather cumbersome modifications to take this into account (similarly to inverting a function). In practice, in all our domains there are several actions that produce the same videos (as the pixel resolution is quite low), and therefore the same intent. Thus we do not believe this to be a strong limitation.
> >
> > ## References
> > Sohl-Dickstein et al. Deep Unsupervised Learning Using Nonequilibrium Thermodynamics. ICML 2015
> >
> > Ho et al. Denoising Diffusion Probabilistic Models. NeurIPS 2020
> >
> > Baker et al. Video pretraining (VPT): Learning to act by watching unlabeled online videos. arXiv:2206.11795, 2022.

---

> > > ### Author Response · Authors · 2022-11-15
> > > **Follow-up**
> > >
> > > Dear Reviewer, we would like to ask if your concerns over the motivation of our method and the distance differences have been addressed, or if there were any other issues that would prevent you from increasing your score. Please let us know, and thank you for your time and feedback.

---

### Author Response · Authors · 2022-12-12
**GAIL Results**

Dear reviewers,

As promised, we performed experiments with GAIL on the particle environment.

We used the GAIL implementation from the same codebase our PPO implementation was based on [1], modified to take state + next state pairs instead of state + action pairs. To add the intent as context to GAIL, we first downscale it from 4096 to 256 using a FC layer and then concatenate it to the state transition. In each experiment, the same trajectories used for $D_{steer}$ were also used as expert trajectories for training the GAIL discriminator.

Our results are as we expected: Except for the case of a single trajectory in the data (the standard GAIL setup of a single task) the training loss of GAIL is significantly outperformed by RL with `STATE-MSE` reward (as defined in Section 5, subsection "Comparison with RL Baselines" in the paper). To further verify this, we experimented with a method that combines the GAIL discriminator with the `STATE-MSE` reward, which outperformed vanilla GAIL, but was still much worse than RL. This is since, as we explained, the dense `STATE-MSE` reward is a very suitable reward signal for this problem, and GAIL does not learn a better reward.

In all cases, the test performance of GAIL (and of RL) is significantly outperformed by IT-IN.

This is not surprising - as we explained, accurately identifying the patterns in the high dimensional intent vector is difficult for RL based methods.

The results are summarized in the following tables. The evaluation protocol is the same as in the paper - sum over all steps of Euclidean distances between agent state variables, of reference and policy trajectories. Values reported are mean and standard deviation over 3 seeds.

### **Training**
**Policy performance on the trajectories from $D_{Steer}$**

| $\|D_{Steer}\|$ | IT-IN | PPO w. ObsMSE Reward |  GAIL |GAIL + ObsMSE Reward |
| ------------ | --------------- | --------------- | --------------- | --------------- |
| 1            | $1.94 \pm 0.07$ | $0.19 \pm 0.09$ | $0.65 \pm 0.71$ | $0.11 \pm 0.07$ |
| 2            | N/A             | N/A             | $2.28 \pm 0.59$ | $0.86 \pm 0.76$ |
| 5            | N/A             | N/A             | $3.51 \pm 0.43$ | $1.13 \pm 0.45$ |
| 10           | $3.71 \pm 0.00$ | $0.48 \pm 0.12$ | $5.25 \pm 0.35$ | $1.38 \pm 0.09$ |
| 50           | $2.62 \pm 0.05$ | $0.75 \pm 0.15$ | $8.92 \pm 1.48$ | $2.83 \pm 0.34$ |
| 100          | $2.78 \pm 0.02$ | $0.98 \pm 0.06$ | $8.66 \pm 0.49$ | $3.37 \pm 0.24$ |
| 500          | $2.66 \pm 0.03$ | $2.27 \pm 0.09$ | N/A             | N/A             |
| 2000         | $2.6 \pm 0.0$   | $3.38 \pm 0.07$ | N/A             | N/A             |


### **Test**
**Policy performance on a held-out test set of 2000 trajectories**

| $\|D_{Steer}\|$ | IT-IN | PPO w. ObsMSE Reward | GAIL | GAIL + ObsMSE Reward |
| ------------ | --------------- | --------------- | --------------- | --------------- |
| 1            | $5.5 \pm 0.5$   | $8.78 \pm 0.04$ | $9.48 \pm 0.21$ | $9.55 \pm 0.09$ |
| 2            | N/A             | N/A             | $9.63 \pm 0.22$ | $9.45 \pm 0.09$ |
| 5            | N/A             | N/A             | $8.5 \pm 0.6$   | $8.02 \pm 0.16$ |
| 10           | $4.7 \pm 0.42$  | $6.84 \pm 0.03$ | $7.89 \pm 0.76$ | $7.67 \pm 0.3$  |
| 50           | $3.1 \pm 0.01$  | $5.18 \pm 0.03$ | $7.69 \pm 0.5$  | $5.67 \pm 0.14$ |
| 100          | $2.8 \pm 0.06$  | $4.71 \pm 0.0$  | $7.85 \pm 0.71$ | $5.34 \pm 0.14$ |
| 500          | $2.6 \pm 0.02$  | $3.98 \pm 0.07$ | N/A             | N/A             |
| 2000         | $2.6 \pm 0.03$  | $3.82 \pm 0.04$ | N/A             | N/A             |

[1] Ilya Kostrikov. Pytorch implementations of reinforcement learning algorithms. https://github.com/ikostrikov/pytorch-a2c-ppo-acktr-gail, 2018.

---

### Decision · Program_Chairs · 2023-01-20

**Decision:**

Reject

**Justification For Why Not Higher Score:**

The work is interesting, but the paper needs to better elucidate the contributions relative to the large body of work in this area. The authors' discussion helped as noted by one of the reviewers increasing their score. The more negative of the three reviewers did engage in a discussion with the authors and the AC, albeit very late, however the reviewer felt that the author response hindered rather than helped their understanding of the paper, which the reviewer felt was fairly solid.

**Justification For Why Not Lower Score:**

N/A

**Metareview: Summary, Strengths And Weaknesses:**

The paper describes a formulation of learning from visual demonstrations (i.e., without actions) that uses the assumed invertibility of the dynamics to identify the latent actions corresponding to the observed state trajectory. This inversion is achieved via the proposed "iterative inversion" strategy that, as the name suggests, learns the inverse mapping in an iterative fashion in a manner that handles distribution shift. The method is theoretically shown to converge to the correct mapping under certain conditions. Iterative inversion is then used to learn to imitate trajectories provided via visual demonstrations. Experiments on toy domains as well as standard simulation-based control environments demonstrate that the method performs favorably to reward-based (RL) methods.

The problem of learning control policies from visual demonstrations that lack the underlying actions, is of significant practical importance and of interest to many in the (robot) learning community. The reviewers agree that the proposed iterative inversion method is principled. The reviewers particularly appreciate the method for learning the inverse model from a full trajectory as opposed to state pairs, as well as the way in which steering is used to learn this inversion in the face of distribution shift. In their initial reviews, the reviewers raised concerns about the need to more clearly place this work in the context of the large body of work that uses inverse models to relate observations to actions. This includes both an expanded qualitative discussion of related work as well as comparisons to additional baselines. Another concern is that the theoretical results rely upon fairly strict assumptions that seem difficult to satisfy in practice. That said, the empirical results are encouraging, but it would be nice to connect the theory to practice. The authors clearly made a concerted effort to respond to the reviewers' concerns, which included the addition of new experimental results. Key to this response is a discussion of the method's novelty compared to existing work, namely with regards to the use of steering to mitigate distribution shift as noted above. Over the course of the review process, the authors point out what they believe to be a fundamental misunderstanding on the part of Reviewer 9eeh regarding the core contributions of the paper. In discussing the issue with the reviewer, they commented that they feel that they have a good grasp of the paper as presented and that the authors' responses do not clarify the reviewer's understanding of the work and its relationship to DAgger. It is unfortunate that this could not have been resolved sooner.

The AC recognizes the novelty of the idea that one can mitigate distribution shift using steering, which is unique compared to typical approaches to policy/control learning. Clarifying the significance of these contributions relative to existing work, incorporating a more thorough comparison to imitation and inverse RL baselines as appropriate, and revisiting the theoretical analysis in light of the empirical results would go a long way in strengthening the paper.

**Summary Of Ac-Reviewer Meeting:**

N/A